# SOFAR: Language-Grounded Orientation Bridges Spatial Reasoning and Object Manipulation

**Zekun Qi**[13*]  **Wenyao Zhang**[237*]  **Yufei Ding**[34*]  **Runpei Dong**[5]  **Xinqiang Yu**[3]
**Jingwen Li**[4]  **Lingyun Xu**[4]  **Baoyu Li**[5]  **Xialin He**[5]  **Guofan Fan**[1]  **Jiazhao Zhang**[3]  **Jiawei He**[3]
**Jiayuan Gu**[6]  **Xin Jin**[7]  **Kaisheng Ma**[1]  **Zhizheng Zhang**[3‡]  **He Wang**[34‡]  **Li Yi**[18‡]

[1]Tsinghua University  [2]Shanghai Jiao Tong University  [3]Galbot  [4]Peking University  [5]UIUC
[6]ShanghaiTech University  [7]Eastern Institute of Technology  [8]Shanghai Qi Zhi Institute

🏠 Project Page  ⦿ GitHub Code  🤗 HuggingFace

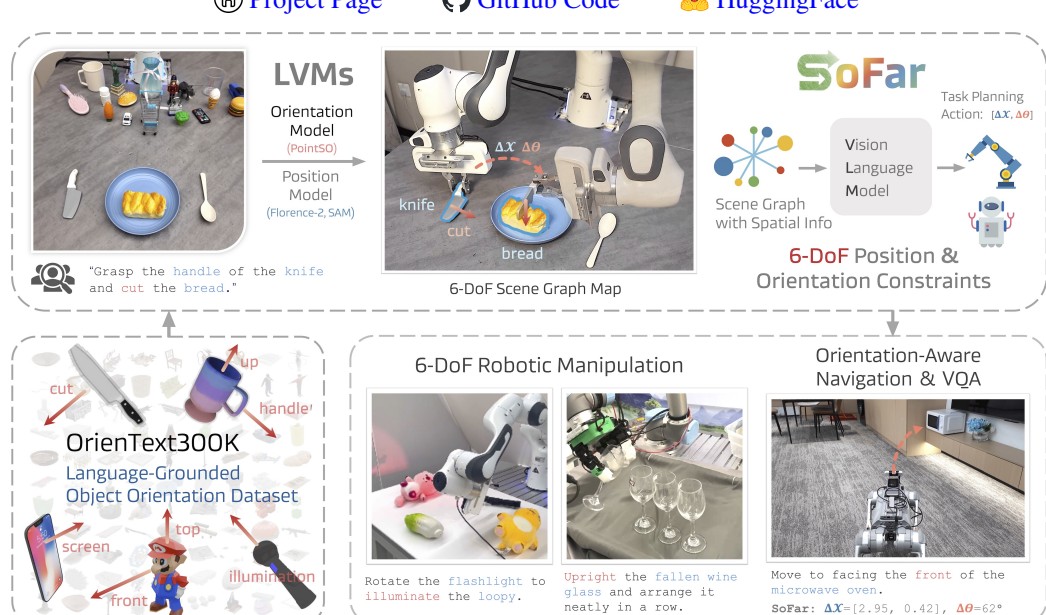

Figure 1: We introduce the concept of *Semantic Orientation*, which refers to natural language-grounded object orientations, such as the "cutting" direction of a knife or the "handle" direction of a cup. To support this, we construct OrienText300K, a large-scale object-text-orientation pairs dataset.

## Abstract

While spatial reasoning has made progress in object localization relationships, it often overlooks object orientation—a key factor in 6-DoF fine-grained manipulation. Traditional pose representations rely on pre-defined frames or templates, limiting generalization and semantic grounding. In this paper, we introduce the concept of semantic orientation, which defines object orientations using natural language in a reference-frame-free manner (*e.g.*, the "plug-in" direction of a USB or the "handle" direction of a cup). To support this, we construct OrienText300K, a large-scale dataset of 3D objects annotated with semantic orientations, and develop PointSO, a general model for zero-shot semantic orientation prediction. By integrating semantic orientation into VLM agents, our SOFAR framework enables 6-DoF spatial reasoning and generates robotic actions. Extensive experiments demonstrated the effectiveness and generalization of our SOFAR, *e.g.*, zero-shot 48.7% successful rate on Open6DOR and zero-shot 74.9% successful rate on SIMPLER-Env.

*Equal contribution. ‡Corresponding author.

39th Conference on Neural Information Processing Systems (NeurIPS 2025).

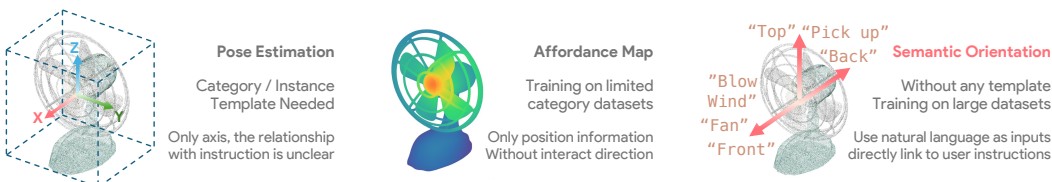

Figure 2: **Representation comparison between semantic orientation and others**.

# 1  Introduction

We observe that current VLMs struggle with understanding object **orientation**, making them insufficient for 6-DoF robot manipulation planning. Consider some everyday scenarios: cutting bread in half with a knife, righting a tilted wine glass, or plugging a cord into a power strip. Previous approaches [10, 12, 8] primarily focused on understanding "*where are the knife and wine glass*" while ignoring their orientations—such as the "blade direction" of the knife and the "up direction" of the glass. This oversight makes it challenging to accomplish these 6-DoF manipulation tasks.

More importantly, different orientations of an object hold varying semantic significance. The capability of connecting specific orientations to their semantic meanings is essential for language-guided robot manipulations. For example, inserting a pen into a pen holder requires aligning the pen tip with the direction of the pen holder's opening; righting a wine glass necessitates aligning the glass's top with the z-axis in the world coordinate frame; and plugging into a power strip involves understanding the "insertion" direction, which is perpendicular to the power strip's surface. However, translating a specific language description into a desired orientation is challenging for existing VLMs.

To move forward, we introduce *language-grounded orientation that bridges spatial reasoning and object manipulation*, characterized by the following:

- **From Position Awareness to Orientation Awareness.** While prior works [10, 12, 8] emphasize position relationship, orientation understanding is equally critical for defining the full 6-DoF of object pose or end-effector poses [16, 120, 124, 60]. Orientation awareness involves understanding object orientations and their relationships in the open world, enabling robots to complete tasks requiring precise alignment and rearrangement.

- **From Orientation to Semantic Orientation.** Traditional orientation, defined relative to a base frame or template model [104, 58, 120, 16], is insufficient for open-world manipulation guided by language instructions [108, 49]. We introduce semantic orientation, linking orientational vectors of an object to open-vocabulary prompts (*e.g.*, the "handle" direction of a knife or "plug-in" direction of a USB). This bridges geometric reasoning with functional semantics, enabling robots to interpret task-specific orientation changes.

Achieving such open-world orientation understanding requires rich world knowledge. To this end, we design both the model architecture and the dataset accordingly. We propose **PointSO**, a generalizable cross-modal 3D Transformer [114, 26, 89, 91] for semantic orientation prediction. To train it at scale, we construct **OrienText300K**, a large-scale dataset comprising over 350K 3D models with diverse orientation-text pairs. These annotations are from Objaverse [20] and generated automatically by prompting GPT-4o [48] with rich semantic queries covering both intra-object spatial reasoning and inter-object manipulation contexts—eliminating the need for costly robot-collected data.

To enable comprehensive spatial reasoning, we develop **SoFar**, an integrated system that combines PointSO with foundation models such as SAM [57]. Given an RGB-D input, SAM segments the scene, and PointSO estimates object orientations to build an orientation-aware 3D scene graph. The graph together with the image is fed into a VLM to generate chain-of-thought [119] spatial reasoning, supporting both positional and orientational planning for downstream robotic manipulation.

In addition, we introduce Open6DOR V2, a large-scale benchmark for 6-DoF object rearrangement in simulation, which supports both open-loop and closed-loop control. Our method significantly outperforms state-of-the-art VLMs and VLA models—even those trained on expensive robot trajectories—across both simulated and real-world tasks. We also introduce 6-DoF SpatialBench, a new spatial visual-question-answering benchmark to rigorously assess orientation-aware reasoning.

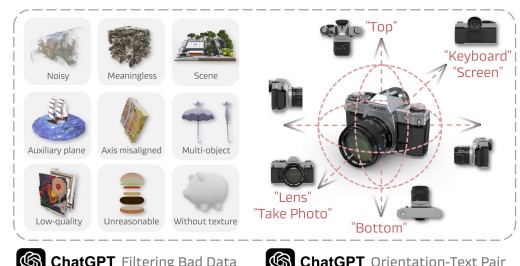 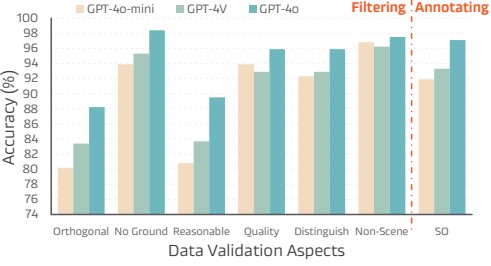


(a) Data Construction of OrienText300K.     (b) Data filtering and annotating accuracy.


Figure 3: **Visualization of OrienText300K data construction and validation results**.

In summary, we propose **Semantic Orientation** as a new representation that bridges spatial reasoning and robotic manipulation, enabling open-vocabulary, template-free orientation understanding for unseen objects. We introduce **OrienText300K**, a large-scale dataset including 350K diverse objects & orientations and 8M images through careful filtering and annotating. We develop the SOFAR system, which enhances spatial reasoning with 6-DoF scene graph and achieves SOTA performance on Open6DOR, SimplerEnv, and generalizes across embodiments (*e.g.*, grippers, suction cups, dexterous hands) and tasks (*e.g.*, manipulation, navigation, VQA) without any task-specific fine-tuning. Finally, we present two new benchmarks, **Open6DOR V2** and **6-DoF SpatialBench**, to evaluate 6-DoF rearrangement and spatial reasoning.

## 2 Semantic Orientation: Connecting Language and Object Orientation

### 2.1 Definition of Semantic Orientation

Traditionally, object orientation is defined within a reference frame using quaternions or Euler angles to describe relative rotations. However, in interactive tasks, orientations often carry semantic meaning. Humans naturally interpret orientation in a semantic, reference-free manner. For example, plugging in a charger involves aligning the metal prongs with the socket's opening direction—a semantically grounded alignment. Motivated by this, we define an object's *Semantic Orientation* as a unit vector that captures the direction corresponding to a given language description. Formally, for an object $X$ and a description $\ell$, the semantic orientation $\mathbf{s}_\ell^X \in S(2)$ is defined as:

$$\mathbf{s}_\ell^X = \mathcal{F}(X, \ell). \tag{1}$$

Here, $\ell$ is open-vocabulary phrase referring to general directions (*e.g.*, *front*, *top*), object parts (*e.g.*, *handle*, *cap*), or interactions (*e.g.*, *pour out*, *plug-in*). An object $X$ can be associated with multiple semantic orientations by varying the language input, forming a set $S_X = \{\mathbf{s}_{\ell_1}^X, \mathbf{s}_{\ell_2}^X, \ldots, \mathbf{s}_{\ell_n}^X\}$. These orientations provide a semantic basis for describing and transforming the object's rotation.

### 2.2 OrienText300K: Orientation-Text Paired Data at Scale

Our goal is to develop an *orientation model* capable of identifying semantic orientations in open-world settings using large-scale 3D data. To support this, we introduce OrienText300K, a curated dataset of 3D models annotated with diverse language-guided orientation labels. The dataset is constructed from Objaverse [20], which contains approximately 800K Internet-sourced 3D models across a wide range of categories. Since the raw data includes noisy annotations and low-quality samples, we apply a rigorous filtering process. Using Blender, we render over 8M high-quality images under carefully designed lighting conditions to ensure fidelity for training.

**Data Filtering** To ensure high-quality data for generating semantic orientation annotations, we apply a dedicated filtering strategy that retains only the samples meeting the following six criteria. ❶ Standard orthogonal view only. Samples in random views will be filtered. ❷ Clean objects without the ground for auxiliary visualization. ❸ Reasonable objects that have sufficient spatial reasoning potentials. ❹ High-quality objects. Blurry and wrong samples are filtered. ❺ Distinguishable objects. Abstract and meaningless objects are filtered. ❻ Non-scene objects for object-centric understanding.

However, it is non-trivial to conduct filtering on such big data using manual labor. Inspired by recent works showing large VLMs are human-aligned judgers [147, 121, 85], we employ GPT-4o [48] by prompting requirements above. To be specific, the multi-view images of 3D objects are concatenated together with our designed prompts into GPT-4o, and GPT-4o will decide whether samples should be filtered. The filtered dataset yields 350K+ clean samples, significantly reducing data noise.

**Data Annotation** As mentioned in the introduction, VLMs struggle to produce accurate object orientation values, which presents a significant challenge for data generation. Fortunately, VLMs are powerful discriminators capable of distinguishing between different views through multimodal understanding. We believe that the initial stage of data cleaning effectively removed a large amount of misaligned data, leaving behind a set of properly aligned instances capable of producing *standard* orthogonal views. We then leverage GPT-4o to interpret the semantic content across six views and generate semantic-view pairs accordingly. Throughout the annotation process, both human modelers in Objaverse and ChatGPT serve as our annotators, supplying the necessary knowledge to produce both view-aligned data and semantically grounded annotations.

**Quality Validation** To validate annotation quality, we construct a validation set containing 208 samples with manually labeled filtering criteria and semantic orientation labels, respectively. From Fig. 3b, we observe that GPT-4o achieves an average accuracy of 88.3% and 97.1% accuracy on filtering and annotating, respectively. This provides a quality guarantee of our OrienText300K.

## 2.3 PointSO: A Cross-Modal 3D Transformer for Semantic Orientation Prediction

We introduce PointSO, a plain Transformer-based architecture [114] with cross-modal 3D-language fusion as our orientation model. As illustrated in Fig. 4, PointSO takes the object's 3D point clouds and a language description as inputs, and predicts the corresponding semantic orientation.

**3D and Language Embeddings** Given an object's point cloud $X = \{\mathbf{x}_i \in \mathbb{R}^3 | i = 1, 2, \ldots, N\}$ with $N$ 3D points defined in (x, y, z) Cartesian space, and an arbitrary language description $\ell$, we first embed both into discrete token embeddings. For the 3D point clouds, we follow [26, 136, 89] to first sample $N_s$ seed points using farthest point sampling (FPS) and then group inputs with KNN for point feature embedding with a local geometric extraction network such as lightweight PointNet [86, 87]. An MLP head is used which maps a special [CLS] token [28] to a predicted direction. As for the language inputs, we adopt CLIP [97] and use the global token as cross-modal fusion inputs.

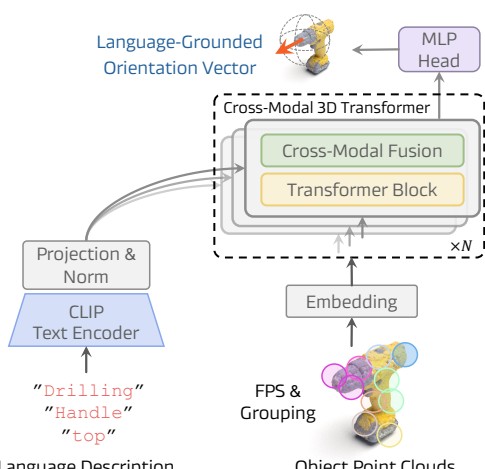

Figure 4: **PointSO model architecture**.

**Cross-Modal Fusion** We perform cross-modal fusion by injecting global text features into each layer of the 3D Transformer using a simple yet effective strategy: adding the text token to every point token. While other fusion methods such as cross-attention, adapters, or concatenation along spatial or channel dimensions are possible, we empirically find that token-wise addition performs best (see Appendix C.3). This effectiveness may stem from the short language inputs, where summation helps reinforce their influence across layers.

**Optimization** Let $\mathcal{F}_{\text{SO}}$ represent the PointSO model parameterized by $\theta_{\text{SO}}$ (the CLIP is kept frozen and thus its parameters are not included). Given every object point cloud $X_i \in \mathcal{D}_{\text{OrienText300K}}$ in the OrienText300K dataset, where each object is labeled with a language set $L_i = \{\ell^i_j, j = 1, 2, \ldots, Q\}$ and the corresponding ground truth semantic orientation set, $S_i = \{\mathbf{s}^i_j, j = 1, 2, \ldots, Q\}$. The optimization is to minimize the negative cosine similarity $\mathcal{L}_{\cos}(\mathbf{v}, \mathbf{k}) = 1 - \frac{\mathbf{v} \cdot \mathbf{k}}{\|\mathbf{v}\| \cdot \|\mathbf{k}\|}$ between predicted and the ground truth semantic orientations:

$$\min_{\theta_{\text{SO}}} \sum_{X_i \in \mathcal{D}_{\text{OrienText300K}}} \sum_{\ell^i_j \in L_i} \mathcal{L}_{\cos}\Big(\mathcal{F}_{\text{SO}}(X_i, \ell^i_j), \mathbf{s}^i_j\Big). \tag{2}$$

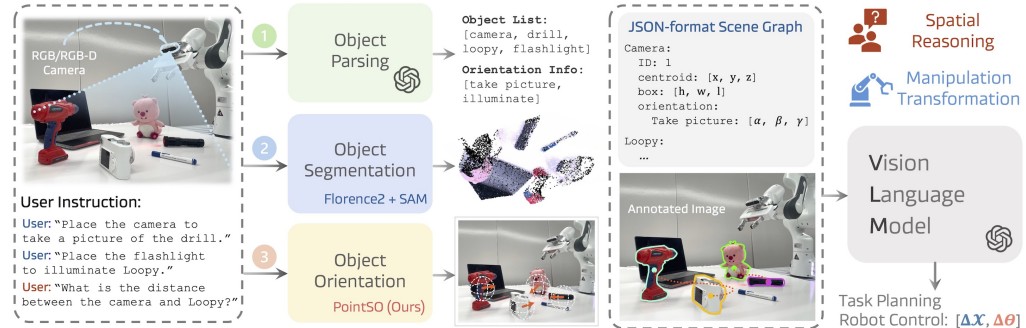

Figure 5: **Overview of SOFAR system.** Given RGB-D images and language instructions, SOFAR first leverages a VLM to identify relevant object phrases and semantic orientations. Then utilizes foundation models Florence-2 [125], SAM [57], and our PointSO for object segmentation and semantic orientation estimation. This information forms a 6-DoF scene graph, which the VLM uses alongside the RGB image to perform spatial understanding tasks or generate manipulation actions.

# 3 SOFAR: Semantic Orientation Bridges Spatial Reasoning and Object Manipulation

Our proposed PointSO model now paves the way for off-the-shelf object-centric spatial orientation understanding. However, it remains challenging to extend such object-centric spatial understanding for scene-level spatial reasoning both in the digital world (*e.g.*, 6-DoF visual question answering) and in the physical world (*e.g.*, robot manipulations). To bridge this gap, we build an integrated reasoning system where a powerful VLM acts as an agent and reasons about the scene while communicating with off-the-shelf models including PointSO and SAM [57]. Fig. 5 illustrates an overview of our proposed framework, aiming at **S**emantic **O**rientation **F**or **A**utonomous **R**obots (**SOFAR**).

## 3.1 Scene Graph with 6-DoF Information

To integrate both the positional & orientational interaction relationships of objects, we use a scene graph with 6-DoF information to represent the environment.

**Position & Orientation Information Extraction** Given a language query $\mathcal{Q}$, we first prompt a vision-language model $\mathcal{F}_{\text{VLM}}$ to extract a task-relevant set of object phrases $\mathcal{P} = \{p_i \,|\, i = 1, 2, \ldots, M\}$. Each phrase $p_i$ represents a language description of an object relevant to $\mathcal{Q}$. Using the SAM [57] & Florence-2 [125], we perform language-conditioned segmentation to obtain a corresponding object set $\mathcal{X} = \{X_i \,|\, i = 1, 2, \ldots, M\}$, where $X_i$ is the 3D point cloud of the $i$-th object. Each object is assigned a unique ID for use in Set-of-Mark (SoM) prompting [129]. We then prompt the VLM to generate a set of task-specific orientation descriptions $L_i$ for related objects, and use pretrained PointSO to infer their semantic orientations, resulting in a semantic orientation set $S_i$.

**6-DoF Scene Graph** From the segmented object set $\mathcal{X}$, we construct an 6-DoF scene graph $\mathcal{G} = (\mathbf{V}, \mathbf{E})$ with $M$ nodes. Each node $\mathbf{o}_i \in \mathbf{V}$ encodes the following semantic and spatial attributes: ❶ object phrase $p_i$ with a unique instance ID; ❷ 3D position $\mathbf{c}_i = (x, y, z) \in \mathbb{R}^3$ from the object's centroid; ❸ bounding box size $\mathbf{b}_i = (h, w, l) \in \mathbb{R}^3$; ❹ semantic orientation set $S_i$ along with its corresponding description set $L_i$. Each edge $\mathbf{e}_{ij} \in \mathbf{E}$ represents the relative translation and size ratio between two connected objects $\mathbf{o}_i$ and $\mathbf{o}_j$.

## 3.2 Spatial-Aware Task Reasoning

We encode the 6-DoF scene graph $\mathcal{G}$ into descriptive language and input it to the VLM alongside the RGB image $I$ and query $\mathcal{Q}$. This enriched spatial representation enables the VLM to perform accurate spatial reasoning by leveraging its visual and linguistic understanding.

**Chain-of-Thought Spatial Reasoning** Most robot manipulation tasks involving rigid objects can be abstracted as applying transformations to adjust their position and orientation. To guide the VLM in generating such transformations from language instructions, we adopt a CoT reasoning process [119] that decomposes the reasoning into three steps: (i) analyzing the scene with the query $\mathcal{Q}$ and object

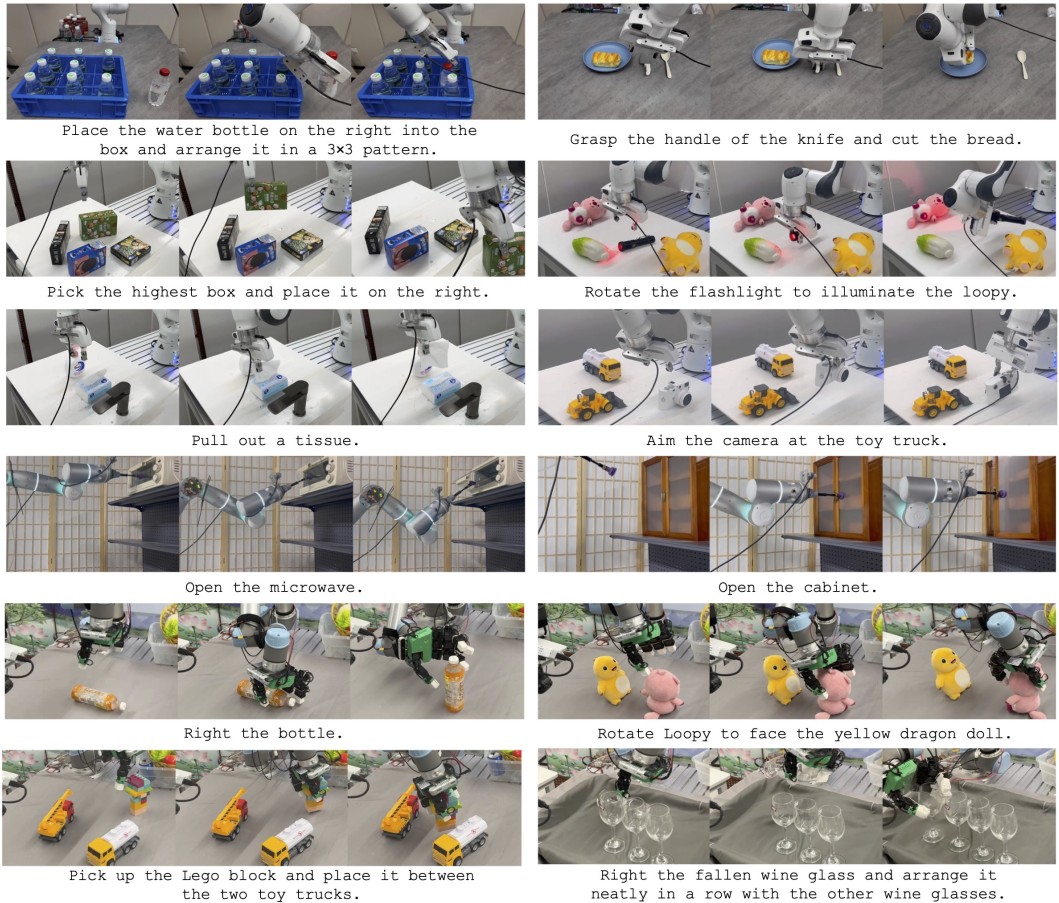

Figure 6: **Qualitative results** of real world language-grounded manipulation. SOFAR can generalize across various **embodiments, tasks and environments**.

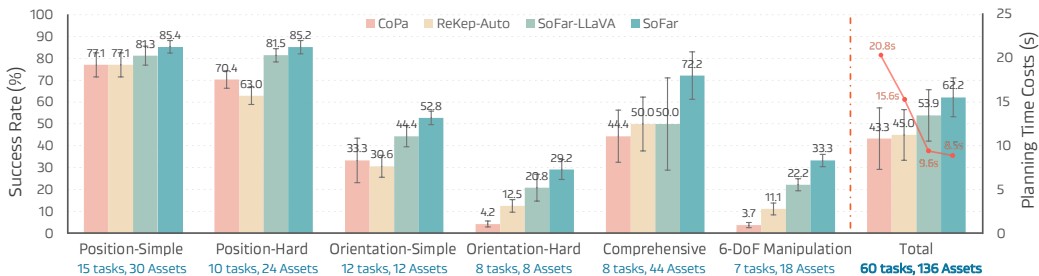

Figure 7: **Quantitative evaluation** of zero-shot real-world language-grounded rearrangement. We design **60** diverse real-world tasks involving over **100** diverse objects (detailed in Table 13).

nodes $\mathbf{V}$; (ii) computing the desired position and orientation of the target object; (iii) predicting the target position $\tilde{\mathbf{c}}_i$ and semantic orientation set $\tilde{S}_i$ for each object. Given the initial state $\mathbf{c}_i$ and $S_i$, the full 6-DoF transformation $\mathbf{P}_i$ is computed. Specifically, translation is obtained by $\mathbf{t}_i = \tilde{\mathbf{c}}_i - \mathbf{c}_i$, and rotation $\mathbf{R}_i$ is estimated from $S_i$ and $\tilde{S}_i$ using the Kabsch-Umeyama algorithm [52, 53, 112].

**Low-Level Motion Execution** Following CoPa [44], we integrate task-specific grasping and motion planning. Object or part segmentation is performed using Florence-2 [125] and SAM [57], followed by grasp candidate generation via GSNet [33]. The optimal grasp is selected by considering both grasp quality and heuristics. Based on instruction, SOFAR predicts the object's translation and rotation, defining the transformation from grasp to placement. We employ OMPL [103] to generate a collision-free trajectory, initializing joint positions at the midpoint to ensure smooth and safe motion.

Table 1: **6-DoF object rearrangement evaluation** on Open6DOR [25].

| Method | Position Track | | | Rotation Track | | | | 6-DoF Track | | | Time Cost (s) |
|---|---|---|---|---|---|---|---|---|---|---|---|
| | Level 0 | Level 1 | Overall | Level 0 | Level 1 | Level 2 | Overall | Position | Rotation | Overall | |
| *Perception Tasks on Issac Sim [80] (Open6DOR V1 Setting)* | | | | | | | | | | | |
| GPT-4V [81] | 46.8 | 39.1 | 45.2 | 9.1 | 6.9 | 11.7 | 9.2 | - | - | - | - |
| Dream2Real [54] | 17.2 | 11.0 | 15.9 | 37.3 | 27.6 | 26.2 | 31.3 | 26.2 | 18.7 | 13.5 | 358.3s |
| VoxPoser [46] | 35.6 | 21.7 | 32.6 | - | - | - | - | - | - | - | - |
| Open6DOR-GPT [25] | 78.6 | 60.3 | 74.9 | 45.7 | 32.5 | 49.8 | 41.1 | 84.8 | 40.0 | 35.6 | 126.3 s |
| SOFAR-LLaVA | 86.3 | 57.9 | 78.7 | 62.5 | 30.2 | 67.1 | 48.6 | 83.0 | 48.2 | 40.3 | 9.6s |
| SOFAR | **96.0** | **81.5** | **93.0** | **68.6** | **42.2** | **70.1** | **57.0** | **92.7** | **52.7** | **48.7** | **8.5s** |
| *Execution Tasks on Libero [64] (Open6DOR V2 Setting)* | | | | | | | | | | | |
| Octo [107] | 51.2 | 32.1 | 47.2 | 10.7 | 18.3 | 29.9 | 17.2 | 45.6 | 8.0 | 8.0 | - |
| OpenVLA [56] | 51.6 | 32.4 | 47.6 | 11.0 | 18.5 | 30.6 | 17.6 | 46.2 | 8.2 | 8.2 | - |
| SOFAR | **72.1** | **47.6** | **67.0** | **28.3** | **18.3** | **34.7** | **25.7** | **63.7** | **25.6** | **18.4** | **40s** |

# 4 Experiments

## 4.1 Real-world Language-Grounded Object Manipulation

**Tasks and Evaluations** We construct 60 real-world tasks involving over 100 objects, following the Open6DOR benchmark [25]. The tasks are divided into three tracks—position, orientation, and comprehensive & 6-DoF—each with simple and hard variants. The position track assesses spatial reasoning from basic (*e.g.*, front/back/left/right) to more complex relations (*e.g.*, between/center/custom). The orientation track includes part-level orientation in the simple setting, and fine-grained angle estimation in the hard setting. The comprehensive and 6-DoF tracks evaluate complex instruction understanding and simultaneous control over position and orientation. Each task is repeated three times to ensure statistical robustness. More details and visualizations are available in Appendix D.1.

**Results** As shown in Fig. 7, SOFAR consistently outperforms baselines across all tracks, especially on orientation and 6-DoF tasks, while maintaining low planning overhead. We also demonstrate SOFAR's embodiment generality with different end-effectors, including dexterous hands and suction cups, as illustrated in Fig. 6. Additional robot setups and generalization results are provided in Appendix A.

## 4.2 Semantic Orientation Prediction

Using free-text descriptions to extract semantic orientations from object point clouds is challenging. In Objaverse [20], we manually annotate 128 diverse objects and construct the OrienText300K val split to evaluate the directional prediction accuracy of PointSO. We train different model variants on OrienText300K, and the results in Table 2 report performance across different angular thresholds ranging from 45° to 5°. PointSO still has an accuracy rate of 60% even under a 5° threshold.

In the real world, obtaining complete object point clouds is often difficult. To evaluate the robustness of PointSO under such conditions, we introduce three types of input perturbations: random rotations, partial single-sided observations, and Gaussian noise. As reported in Table 3, the accuracy at the 45° threshold reflects the model's resilience to these corruptions.

Table 2: **Semantic Orientation evaluation** on OrienText300K validation split.

| Method | 45° | 30° | 15° | 5° | Avg. |
|---|---|---|---|---|---|
| PointSO-S | 77.34 | 74.22 | 67.97 | 60.94 | 70.12 |
| **PointSO-B** | **79.69** | **77.34** | **70.31** | **62.50** | **72.46** |
| **PointSO-L** | **81.25** | **78.13** | **72.66** | **65.63** | **74.42** |

Table 3: **Semantic Orientation evaluation** of *robustness*. `Single-View`: randomly select a camera viewpoint within the unit sphere and generate a single FoV viewpoint in polar coordinates. `Jitter`: Gaussian noise $\epsilon \sim \mathcal{N}(0, \sigma^2)$, $\sigma = 0.01$. `Rotate`: random SO(3) rotation $(\alpha, \beta, \gamma) \sim \mathcal{U}(-\pi, \pi)$. `All`: all corruptions.

| Method | OrienText300K-C Variants | | | |
|---|---|---|---|---|
| | Single-View | Jitter | Rotate | All |
| PointSO-S | 72.66 | 76.56 | 73.43 | 67.19 |
| **PointSO-B** | **75.00** | **78.90** | **75.78** | **71.09** |
| **PointSO-L** | **76.56** | **81.25** | **77.34** | **74.22** |

## 4.3 6-DoF Object Rearrangement Evaluation on Open6DOR V2

To evaluate 6-DoF object rearrangement capabilities, we extend the original Open6DOR benchmark [25], which primarily focuses on final pose estimation, into a more comprehensive setting that

Table 4: **SimplerEnv [62] simulation evaluation results for the Google Robot setup.** We present success rates for the "Variant Aggregation" and "Visual Matching" approaches. Top-1 & Top-2 accuracies are represented using different colors. OXE: Open X-Embodiment dataset [15].

| Google Robot Evaluation Setup | Policy | Training Data | Pick Coke Can | | | | Move Near | Open / Close Drawer | | | Average |
|---|---|---|---|---|---|---|---|---|---|---|---|
| | | | Horizontal Laying | Vertical Laying | Standing | Average | Average | Open | Close | Average | |
| Variant Aggregation | RT-1-X [15] | OXE | 0.569 | 0.204 | 0.698 | 0.490 | 0.323 | 0.069 | 0.519 | 0.294 | 0.397 |
| | RT-2-X [152] | OXE | 0.822 | 0.754 | 0.893 | 0.823 | 0.792 | 0.333 | 0.372 | 0.353 | 0.661 |
| | Octo-Base [107] | OXE | 0.005 | 0.000 | 0.013 | 0.006 | 0.031 | 0.000 | 0.021 | 0.011 | 0.012 |
| | OpenVLA [56] | OXE | 0.711 | 0.271 | 0.653 | 0.545 | 0.477 | 0.158 | 0.195 | 0.177 | 0.411 |
| | **SOFAR** | **Zero-Shot** | **0.861** | **0.960** | **0.901** | **0.907** | 0.740 | 0.200 | 0.394 | 0.297 | **0.676** |
| Visual Matching | RT-1-X [15] | OXE | **0.820** | 0.330 | 0.550 | 0.567 | 0.317 | **0.296** | **0.891** | **0.597** | 0.534 |
| | RT-2-X [152] | OXE | 0.740 | 0.740 | 0.880 | 0.787 | 0.779 | 0.157 | 0.343 | 0.250 | 0.606 |
| | Octo-Base [107] | OXE | 0.210 | 0.210 | 0.090 | 0.170 | 0.042 | 0.009 | 0.444 | 0.227 | 0.168 |
| | OpenVLA [56] | OXE | 0.270 | 0.030 | 0.190 | 0.163 | 0.462 | 0.194 | 0.518 | 0.356 | 0.277 |
| | **SOFAR** | **Zero-Shot** | 0.770 | **1.000** | **1.000** | **0.923** | **0.917** | 0.227 | 0.578 | 0.403 | **0.749** |

Table 5: **SimplerEnv [62] simulation evaluation results for the WidowX + Bridge setup.** We report both the final success rate ("Success") along with partial success (*e.g.*, "Grasp Spoon"). OXE: Open X-Embodiment dataset [15]. Bridge: BridgeData V2 dataset [115] (In domain training).

| Policy | Training Data | Put Spoon on Towel | | Put Carrot on Plate | | Stack Green Block on Yellow Block | | Put Eggplant in Yellow Basket | | Average |
|---|---|---|---|---|---|---|---|---|---|---|
| | | Grasp Spoon | Success | Grasp Carrot | Success | Grasp Green Block | Success | Grasp Eggplant | Success | |
| RT-1-X [6] | OXE | 0.167 | 0.000 | 0.208 | 0.042 | 0.083 | 0.000 | 0.000 | 0.000 | 0.011 |
| Octo-Base [107] | OXE | 0.347 | 0.125 | 0.528 | 0.083 | 0.319 | 0.000 | 0.667 | 0.431 | 0.160 |
| Octo-Small [107] | OXE | 0.778 | 0.472 | 0.278 | 0.097 | 0.403 | 0.042 | 0.875 | 0.569 | 0.300 |
| OpenVLA [56] | OXE | 0.041 | 0.000 | 0.333 | 0.000 | 0.125 | 0.000 | 0.083 | 0.041 | 0.010 |
| RoboVLM [61] | OXE | 0.375 | 0.208 | 0.333 | 0.250 | 0.083 | 0.083 | 0.000 | 0.000 | 0.135 |
| RoboVLM [61] | Bridge | 0.542 | 0.292 | 0.250 | 0.250 | 0.458 | 0.125 | 0.583 | 0.583 | 0.313 |
| SpatialVLA [94] | OXE | 0.250 | 0.208 | 0.417 | 0.208 | 0.583 | 0.250 | 0.792 | 0.708 | 0.344 |
| SpatialVLA [94] | Bridge | 0.208 | 0.167 | 0.292 | 0.250 | 0.625 | 0.292 | **1.000** | **1.000** | 0.427 |
| **SOFAR** | **Zero-Shot** | 0.625 | **0.583** | **0.750** | **0.667** | **0.917** | **0.708** | 0.667 | 0.375 | **0.583** |

includes both perception and execution evaluation. We migrate its scenes into a robosuite-based simulation environment [151], following the task interface defined by LIBERO [64], and name this new benchmark Open6DOR V2. Results are reported in Table 1. For perception tasks, we adopt the original Open6DOR [25] evaluation protocol and compare with the same baselines. SOFAR achieves the best performance, demonstrating strong spatial understanding and zero-shot generalization. For execution tasks, we compare against the pretrained Octo [107] and the LIBERO-finetuned Open-VLA [56], all evaluated in the same robosuite environment to minimize domain shift. While both baselines show limited success due to poor generalizability, SOFAR reaches around 40% success rate using a vanilla execution pipeline. We note that certain objects are intrinsically difficult to manipulate, suggesting the need for more robust policies incorporating prehensile grasping and adaptive strategies to improve performance on Open6DOR V2.

## 4.4 Simulation Object Manipulation Evaluation on SIMPLER [62]

We conduct quantitative evaluations of SOFAR's zero-shot execution performance on Google Robot tasks & Widow-X tasks and compare it to baselines including Octo [107], OpenVLA [56] and more concurrent works [61, 94]. The robot follows the planned trajectory generated by the planning module, as described in Sec. 3.2, to execute the task. Furthermore, leveraging the error detection and re-planning capabilities of VLMs [48, 1], we can make multiple attempts following a single-step execution failure to approximately achieve a closed-loop effect. For fairness, we limit the maximum number of attempts to three. Detailed visualizations and analyses are provided in the Appendix B.5. As shown in Tables 4 and 5, despite the training data for Octo and OpenVLA including Google Robot tasks, SOFAR demonstrates superior zero-shot performance compared to most baselines.

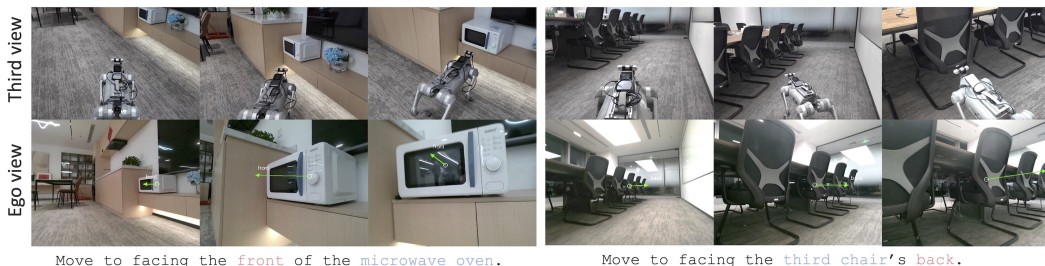

Figure 8: **Real-world orientation-aware navigation.** We present both the third-person view and the egocentric view, annotating the predicted orientation of the interacted objects.

### 4.5 Orientation-Aware Robotic Navigation

In navigation tasks, reaching an object from its functional side is crucial for subsequent manipulation—for example, approaching a microwave from the front to open its door. To support such scenarios, we extend *semantic orientation* to the navigation domain. As shown in Fig. 8, a quadruped robot is tasked with reaching both the correct position and the appropriate facing direction. This orientation-aware constraint enhances the navigation process by ensuring precise alignment with the desired orientation, thereby improving task performance in scenarios where directionality is critical.

### 4.6 Spatial Reasoning Evaluation on 6-DoF SpatialBench

To assess spatial understanding with full 6-DoF awareness, we introduce **6-DoF SpatialBench**, a VQA benchmark designed to evaluate both positional and orientational comprehension. Unlike prior benchmarks [12, 8, 29, 106] that primarily emphasize coarse positional reasoning (*e.g.*, "to the left," "nearest") and often overlook orientation or rely on relative metrics, we provide a more fine-grained evaluation with quantitative annotations. It consists of 223 human-annotated samples, each containing an RGB image and a multiple-choice question with 4 options. The benchmark includes two tracks: position and orientation, covering tasks such as object counting, spatial relations, and object-facing direction. All questions and ground-truth answers are curated through **human annotation**. We evaluate SOFAR on 6-DoF Spatial-Bench against several VLMs and comparable methods as baselines, as presented in Table 6. SOFAR consistently outperforms other methods across both tracks, achieving over 18% improvement.

Table 6: **Spatial reasoning evaluation** on 6-DoF SpatialBench. *Depth-Esti*: Use depth estimation methods such as Metric3D [135] or Moge [117].

| Method | Depth-Esti | Position | | Orientation | | Total |
|---|---|---|---|---|---|---|
| | | rel. | abs. | rel. | abs. | |
| *Blind Evaluation with LLMs* | | | | | | |
| GPT-3.5-Turbo [7] | ✗ | 24.5 | 24.9 | 26.7 | 27.5 | 25.7 |
| GPT-4-Turbo [82] | ✗ | 27.2 | 27.3 | 29.2 | 27.9 | 27.8 |
| *General VLMs* | | | | | | |
| LLaVA-1.5 [68] | ✗ | 30.9 | 24.5 | 28.3 | 25.8 | 27.2 |
| GPT-4o-mini [48] | ✗ | 33.3 | 26.9 | 32.5 | 23.8 | 31.0 |
| GPT-4o [48] | ✗ | 49.4 | 28.4 | 44.2 | 25.8 | 36.2 |
| *VLMs with Spatial Awareness* | | | | | | |
| SpaceLLaVA [10] | ✗ | 32.4 | 30.5 | 30.9 | 24.9 | 28.2 |
| SpaceMantis [10] | ✗ | 33.6 | 29.2 | 27.2 | 25.0 | 28.9 |
| SpatialBot [8] | ✓ | 50.9 | 21.6 | 39.6 | 22.9 | 32.7 |
| RoboPoint [137] | ✗ | 43.8 | 30.8 | 33.8 | 25.8 | 33.5 |
| **SOFAR** | ✓ | **59.6** | **33.8** | **54.6** | **31.3** | **43.9** |

## 5 Limitations & Conclusions

One notable limitation for decoupled systems like SOFAR is that the execution may fail due to a sub-module error, as shown in Appendix B.8, *i.e.*, robots may place target objects with an error transformation because of unstable grasping or inaccurate visual perception. For example, the pen will be placed in an unexpected pose due to the rotation during execution. Future works include integrating scalable data and more advanced models and exploring the potential of combining end-to-end and such decoupled methods, and expanding SOFAR to more applications.

We propose *semantic orientation*, a language-grounded representation that links object orientations with intuitive descriptors (*e.g.*, "plug-in direction"), bridging geometric reasoning and functional semantics. To support this, we construct OrienText300K, a large-scale dataset of 3D models with semantic orientation annotations. Our PointSO model, integrated within the SOFAR system, demonstrates strong performance in both simulated and real-world robotic manipulation tasks.

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

# A Robot Setups

## A.1 Simulation Robot Setups

To ensure fairness, we utilize the same Franka Panda arm for evaluations in both the LIBERO [64] and our Open6DOR V2 benchmarks. For SIMPLER [62], we use the Google Robot and Widow-X exclusively to conduct the baseline experiments, adhering to all configurations outlined in SIMPLER, as presented in Tables 4 and 5.

## A.2 Real World Robot Setups

As for manipulation tasks, in Fig. 9, we perform 6-DoF rearrangement tasks using the Franka Panda equipped with a gripper and the UR robot arm with a LeapHand, while articulated object manipulation is conducted using the Flexiv arm equipped with a suction tool. All the robot arms mount a RealSense D415 camera at their end for image capturing.

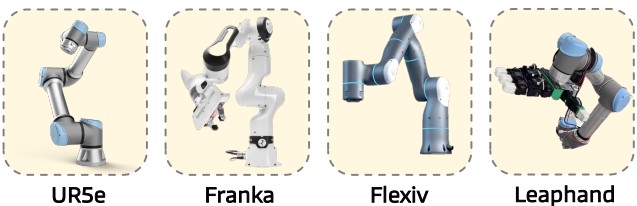

Figure 9: **The robots used in our real-world experiments.**

In Fig. 10, we present the workspace and robotic arm for real-world 6-DoF rearrangement. Unlike Rekep [47], CoPa [44] et al., we utilize only a single RealSense D415 camera. This setup significantly reduces the additional overhead associated with environmental setup and multi-camera calibration, and it is more readily reproducible.

As for navigation tasks, we provide a visualization of our robotic dog in Fig. 11. Following Uni-Navid [139], our robotic dog is Unitree GO2 and we mount a RealSense D455 camera on the head of the robotic dog. Here, we only use the RGB frames with a resolution of $640 \times 480$ in the setting of $90°$ HFOV. We also mount a portable Wi-Fi at the back of the robot dog, which is used to communicate with the remote server (send captured images and receive commands). Unitree GO2 is integrated with a LiDAR-L1, which is only used for local motion planning.

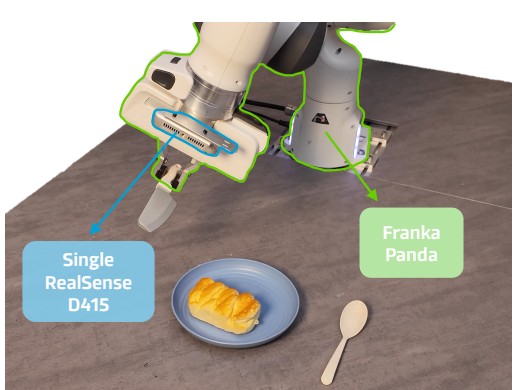

Figure 10: **6-DoF rearrangement robot setup.**

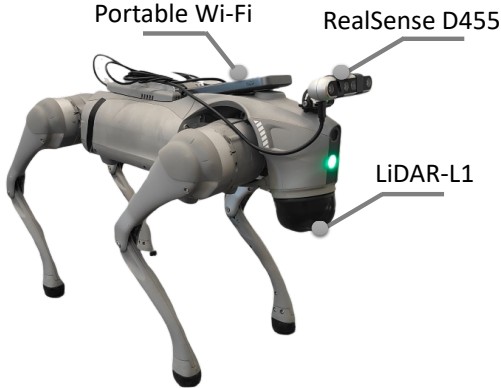

Figure 11: **Navigation robot setup.**

Table 7: **Zeroshot articulate object manipulation evaluation** within the SAPIEN [123] simulator using PartNet-Mobility Dataset. Notably, while the baseline methods use distinct training and testing splits, our model achieves robust performance without fine-tuning on the SAPIEN samples.

| Method | 🔒 | 🚪 | 🖥 | 🗄 | 💻 | 🖊 | 📟 | 🖱 | 📦 | 🗑 | 🍲 | 🖋 | ✂ | 🗄 | 📱 | 🧴 |
|---|---|---|---|---|---|---|---|---|---|---|---|---|---|---|---|---|
| Where2Act [79] | 0.26 | 0.36 | 0.19 | 0.27 | 0.23 | 0.11 | 0.15 | 0.47 | 0.14 | 0.24 | 0.13 | 0.12 | 0.56 | 0.68 | 0.07 | 0.40 |
| UMPNet [128] | 0.46 | 0.43 | 0.15 | 0.28 | 0.54 | 0.32 | 0.28 | 0.56 | 0.44 | 0.40 | 0.10 | 0.23 | 0.18 | 0.54 | 0.20 | 0.42 |
| FlowBot3D [30] | 0.67 | 0.55 | 0.20 | 0.32 | 0.27 | 0.31 | 0.61 | **0.68** | 0.15 | 0.28 | 0.36 | 0.18 | 0.21 | 0.70 | 0.18 | 0.26 |
| Implicit3D [148] | 0.53 | 0.58 | 0.35 | 0.55 | 0.28 | **0.66** | 0.58 | 0.51 | 0.52 | 0.57 | 0.45 | 0.34 | 0.41 | 0.54 | 0.39 | 0.43 |
| ManipLLM [60] | 0.68 | 0.64 | 0.36 | 0.77 | 0.43 | 0.62 | 0.65 | 0.61 | 0.65 | 0.52 | 0.53 | **0.40** | 0.64 | 0.71 | **0.60** | **0.64** |
| **SoFar** | **0.75** | **0.88** | **0.43** | **0.85** | **0.60** | 0.54 | **0.75** | 0.49 | **0.58** | **0.72** | **0.69** | 0.42 | **0.70** | **0.81** | 0.58 | 0.63 |

| Method | 🪜 | 🗄 | 🔨 | 🧴 | AVG | 🚽 | ✂ | 🛋 | 📠 | 🫖 | 🔖 | 📺 | 🎛 | 🚰 | 📱 | AVG |
|---|---|---|---|---|---|---|---|---|---|---|---|---|---|---|---|---|
| Where2Act [79] | 0.13 | 0.18 | 0.13 | 0.40 | 0.26 | 0.18 | 0.35 | 0.38 | 0.28 | 0.05 | 0.21 | 0.17 | 0.20 | 0.15 | 0.15 | 0.21 |
| UMPNet [128] | 0.22 | 0.33 | 0.26 | 0.64 | 0.35 | 0.42 | 0.20 | 0.35 | 0.42 | 0.29 | 0.20 | 0.26 | 0.28 | 0.25 | 0.15 | 0.28 |
| FlowBot3D [30] | 0.17 | 0.53 | 0.29 | 0.42 | 0.37 | 0.23 | 0.10 | 0.60 | 0.39 | 0.27 | 0.42 | 0.28 | 0.51 | 0.13 | 0.23 | 0.32 |
| Implicit3D [148] | 0.27 | 0.65 | 0.20 | 0.33 | 0.46 | 0.45 | 0.17 | 0.80 | 0.53 | 0.15 | 0.69 | 0.41 | 0.31 | 0.30 | 0.31 | 0.41 |
| ManipLLM [60] | **0.41** | **0.75** | 0.44 | 0.67 | 0.59 | 0.38 | 0.22 | 0.81 | **0.86** | 0.38 | **0.85** | 0.42 | **0.83** | 0.26 | 0.38 | 0.54 |
| **SoFar** | 0.35 | 0.68 | **0.62** | **0.73** | **0.64** | **0.68** | **0.45** | **0.90** | 0.77 | **0.55** | 0.79 | **0.48** | 0.80 | **0.56** | **0.44** | **0.64** |

# B  Additional Experiments

## B.1  Articulated Objects Manipulation Evaluation

We further integrate SoFar with articulated object manipulation, as illustrated in Table 7, and evaluate its practicality in robotic manipulation tasks using the PartNet-Mobility Dataset within the SAPIEN [123] simulator. Our experimental setup follows ManipLLM [60], employing the same evaluation metrics. Specifically, we directly utilize the segmentation centers provided by SAM as contact points, leverage PointSO to generate contact directions, and use VLM to determine subsequent motion directions. The results demonstrate significant improvements over the baseline. Notably, our model achieves this performance without dividing the data into training and testing sets, operating instead in a fully zero-shot across most tasks. This underscores the robustness and generalization of our approach.

## B.2  Spatial Reasoning on EmbSpatial-Bench [29]

To further validate the spatial reasoning capabilities of SoFar, we evaluated its performance on the spatial visual-question-answering tasks in EmbSpatial-Bench [29]. As reported in Table 8, our model substantially outperforms all baseline methods, achieving over a 20% improvement in overall performance. This result highlights SoFar's effectiveness in spatial understanding and reasoning within complex visual scenes.

Table 8: Evaluation of EmbSpatial-Bench [29].

| Model | Generation | Likelihood |
|---|---|---|
| BLIP-2 [59] | 37.99 | 35.71 |
| InstructBLIP [18] | 38.85 | 33.41 |
| MiniGPT4 [150] | 23.54 | 31.70 |
| LLaVA-1.6 [67] | 35.19 | 38.84 |
| GPT-4V [81] | 36.07 | - |
| Qwen-VL-Max [3] | 49.11 | - |
| **SoFar** | **70.88** | - |

## B.3  Cross Embodiment Generalization

Our approach determines grasp poses by generating masks and plans the target pose and transformation using our PointSO and large language model. It does not rely on trajectory data specific to any robotic arm, making SoFar embodiment-agnostic. Fig. 6 illustrates the diverse embodiments employed in our real-world experiments. Leveraging the GSNet [116] algorithm based on Leap-Hand [99], we perform 6-DoF object manipulation experiments on dexterous hands. We conduct three position-related and three rotation-related experiments. Leveraging the PointSO and large language models, SoFar is capable of performing complex 6-DoF manipulation tasks, such as "*Upright the fallen wine glass and arrange it neatly in a row with the other wine glasses.*"

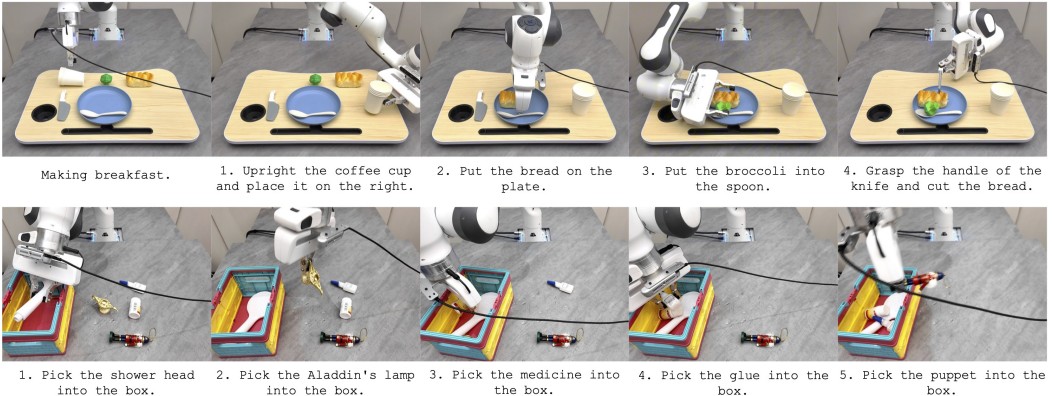

| | | | | |
|---|---|---|---|---|
| Making breakfast. | 1. Upright the coffee cup and place it on the right. | 2. Put the bread on the plate. | 3. Put the broccoli into the spoon. | 4. Grasp the handle of the knife and cut the bread. |
| 1. Pick the shower head into the box. | 2. Pick the Aladdin's lamp into the box. | 3. Pick the medicine into the box. | 4. Pick the glue into the box. | 5. Pick the puppet into the box. |

Figure 12: **Long-horizon object manipulation** experiment of our SOFAR.

## B.4 Long Horizon Object Manipulation Experiment

Fig. 12 illustrates the execution performance of our model on long-horizon tasks. Through the VLM [48, 1], complex instructions such as "making breakfast" and "cleaning up the desktop" can be decomposed into sub-tasks. In the second example, we deliberately chose uncommon objects as assets, such as "Aladdin's lamp" and "puppets", but SOFAR is able to successfully complete all tasks.

## B.5 Close-Loop Execution Experiment

Similar to ReKep [47], SOFAR leverages VLMs [48, 1] to perform long-horizon decomposition of complex tasks and employs dual-system VLMs [48, 1] to determine the success of execution between tasks and subtasks, enabling closed-loop execution. When a discrepancy between the results and expectations is detected, SOFAR re-percepts and re-executes the current subtask. We demonstrate the closed-loop replan capabilities of SOFAR within Simpler-Env [62] in Fig. 13. The instruction for both tasks is "pick the coke can" In Fig. 13

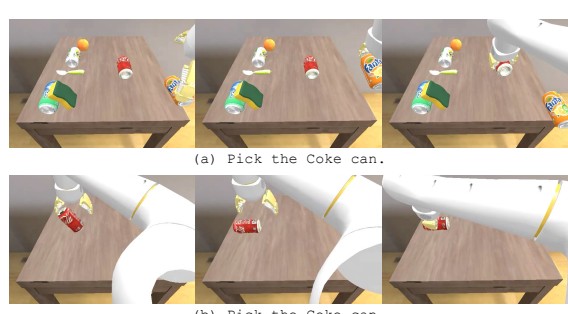

Figure 13: **Close-loop execution of our SOFAR.**

(a), the model initially misidentified the coke can as a Fanta can. After correction by the VLM, the model re-identified and located the correct object. In Fig. 13 (b), the model accidentally knocks over the Coke can during motion due to erroneous motion planning. Subsequently, the model re-plans and successfully achieves the grasp.

## B.6 In the Wild Evaluation of Semantic Orientation

We provide a qualitative demonstration of the accuracy of PointSO under in-the-wild conditions, as shown in Fig. 14, where the predicted Semantic Orientation is marked in the images. We obtained single-sided point clouds by segmenting objects using Florence-2 [125] and SAM [57] and fed them into PointSO. It can be observed that our model achieves good performance across different views, objects, and instructions, which proves the effectiveness and generalization of PointSO.

## B.7 Cross-View Generalization

SOFAR gets point clouds in the world coordinate system using an RGB-D camera to obtain grasping poses, and it is not limited to a fixed camera perspective. In addition, PointSO generates partial point clouds from different perspectives through random camera views to serve as data augmentation for training data, which also generalizes to camera perspectives in the real world. Fig. 15 illustrates SOFAR's generalization capability for 6-DoF object manipulation across different camera poses. It can be observed that whether it's a front view, side view, or ego view, SOFAR can successfully execute the "upright the bottle" instruction.

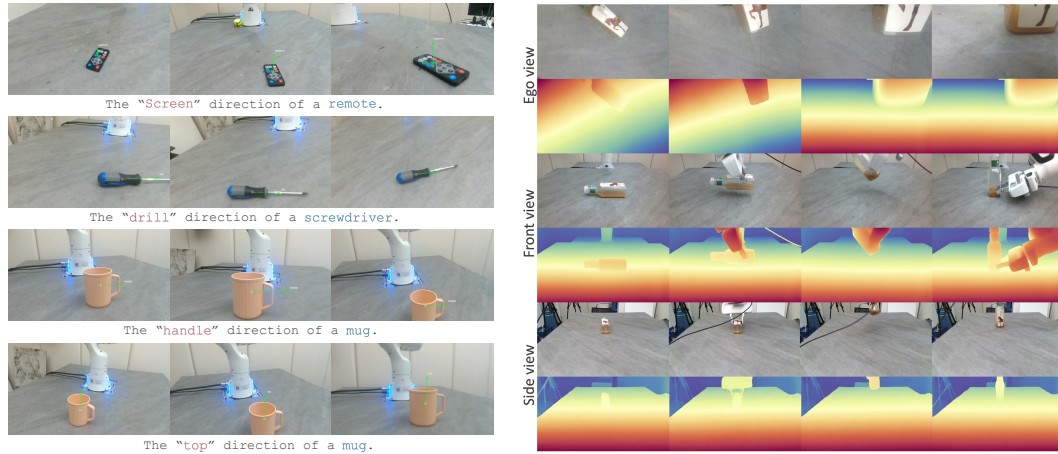

Figure 14: **In-the-wild evaluation** of PointSO.    Figure 15: **Cross view generalization** of SOFAR.

## B.8 Failure Case Distribution Analysis

Based on the failure cases from real-world experiments, we conducted a quantitative analysis of the failure case distribution for SOFAR, with the results shown in Fig. 16. It can be observed that 31% of the failures originated from grasping issues, including objects being too small, inability to generate reasonable grasping poses, and instability after grasping leading to sliding or dropping. Next, 23% were due to incorrect or inaccurate Semantic Orientation prediction. For tasks such as upright or upside - down, highly precise angle estimation (<5°) is required for smooth execution. Object analysis and detection accounted for approximately 20% of the errors. The instability of open-vocabulary detection modules like Florence2 [125] and Grounding DINO [69] often led to incorrect detection of out-of-distribution objects or object parts. In addition, since our Motion Planning did not take into account the working space range of the robotic arm and potential collisions of the manipulated object, occasional deadlocks and collisions occurred during motion. Finally, there were issues with the Task Planning of the VLM [48, 1]. For some complex Orientations, the VLM occasionally failed to infer the required angles and directions to complete the task. Employing a more powerful, thought-enabled VLM [50] might alleviate such errors.

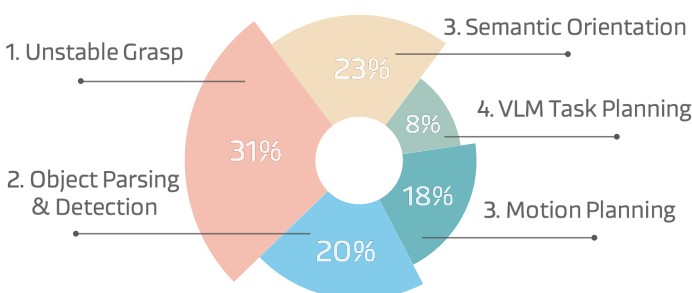

Figure 16: **Failure case distribution analysis** of our SOFAR.

## C  Ablation Study

### C.1  Semantic Orientation Ablation

To demonstrate that the proposed semantic orientation indeed plays a crucial role in robotic tasks—rather than the observed effects being attributable to other factors such as Chain-of-Thought reasoning—we conduct conduct ablation experiments for methodological differences between the baselines, including whether to add semantic orientation in the scene graph and whether to use CoT, as shown in Tab. 9.

Table 9: **Ablation study of composition module** of SOFAR.

| CoT | Orient. | Position Track | | | Rotation Track | | | | 6-DoF Track | | |
|---|---|---|---|---|---|---|---|---|---|---|---|
| | | Level 0 | Level 1 | Overall | Level 0 | Level 1 | Level 2 | Overall | Position | Rotation | Overall |
| ✗ | ✗ | 95.4 | 77.7 | 91.9 | 17.2 | 8.4 | 11.4 | 13.0 | 92.7 | 15.5 | 14.2 |
| ✓ | ✗ | **96.3** | **81.6** | **93.3** | 16.3 | 8.9 | 11.0 | 12.9 | **93.0** | 15.1 | 13.7 |
| ✗ | ✓ | 95.6 | 77.2 | 91.7 | 63.3 | 35.4 | 61.8 | 52.3 | 92.7 | 48.3 | 45.8 |
| ✓ | ✓ | 96.0 | 81.5 | 93.0 | **68.6** | **42.2** | **70.1** | **57.0** | 92.7 | **52.7** | **48.7** |

## C.2 Scaling Law

The scaling capability of models and data is one of the most critical attributes today and a core feature of foundation models [5]. We investigate the performance of PointSO across different data scales, as illustrated in Table 10. We obtain the subset for OrienText300K from Objaverse-LVIS, which consists of approximately 46,000 3D objects with category annotations. The selection was based on the seven criteria mentioned in the main text. Objects meeting all seven criteria formed the strict subset, comprising around 15k objects. When including objects without textures and those of lower quality, the total increases to approximately 26k objects. It can be seen that the increase in data volume is the most significant factor driving the performance improvement of PointSO. It can be anticipated that with further data expansion, such as Objaverse-XL [19], PointSO will achieve better performance.

Table 10: **Data scaling property** of semantic orientation with different training data scales evaluated on OrienText300K validation split. All experiments are conducted with the PointSO-Base variant.

| Data Scale | 45° | 30° | 15° | 5° | Average |
|---|---|---|---|---|---|
| 15K | 57.03 | 46.09 | 39.84 | 27.34 | 42.58 |
| 35K | 61.72 | 53.13 | 43.75 | 30.47 | 47.27 |
| 150K | 76.56 | 72.66 | 66.41 | 56.25 | 67.97 |
| 350K | **79.69** | **77.34** | **70.31** | **62.50** | **72.46** |

## C.3 Cross-Modal Fusion Choices

We further conduct an ablation study on the multi-modal fusion methods in PointSO, testing commonly used feature fusion techniques such as cross-attention, multiplication, addition, and concatenation, as shown in Table 11. The results indicate that simple addition achieves the best performance. This may be attributed to the fact that instructions in the semantic domain are typically composed of short phrases or sentences, and the text CLS token already encodes sufficiently high-level semantic information.

Table 11: **Ablation study of multi-modal fusion** in PointSO. All experiments are conducted with the PointSO-Base variant.

| Fusion Method | 45° | 30° | 15° | 5° | Avg. |
|---|---|---|---|---|---|
| Cross-attn | 74.22 | 70.31 | 63.28 | 57.03 | 66.21 |
| Multiplication | 74.22 | 69.53 | 60.16 | 56.25 | 65.04 |
| Addition | **79.69** | **77.34** | **70.31** | **62.50** | **72.46** |
| Concat | 66.41 | 60.94 | 52.34 | 43.75 | 55.86 |

Table 12: **Ablation study of open vocabulary detection modules** on Open6DOR perception tasks.

| Method | Position Track | | | Rotation Track | | | | 6-DoF Track | | | Time Cost (s) |
|---|---|---|---|---|---|---|---|---|---|---|---|
| | Level 0 | Level 1 | Overall | Level 0 | Level 1 | Level 2 | Overall | Position | Rotation | Overall | |
| YOLO-World [13] | 59.0 | 37.7 | 53.3 | 48.3 | 36.1 | 62.0 | 44.9 | 53.4 | 44.6 | 27.8 | **7.4s** |
| Grounding DINO [69] | 92.2 | 71.5 | 86.7 | 64.7 | 41.1 | 69.8 | 55.5 | 87.2 | 51.6 | 44.6 | 9.2s |
| Florence-2 [125] | **96.0** | **81.5** | **93.0** | **68.6** | **42.2** | **70.1** | **57.0** | **92.7** | **52.7** | **48.7** | 8.5s |

## C.4 Open Vocabulary Object Detection Module

SOFAR utilize an open vocabulary detection foundation model to localize the interacted objects or parts, then generate masks with SAM [57]. Although not the SOTA performance on the COCO benchmark, Florence-2 [125] exhibits remarkable generalization in in-the-wild detection tasks, even in simulator scenarios. Table 12 illustrates the performance of various detection modules in Open6DOR [25] Perception, where Florence-2 achieves the best results and outperforms Grounding DINO [69] and YOLO-World [13].

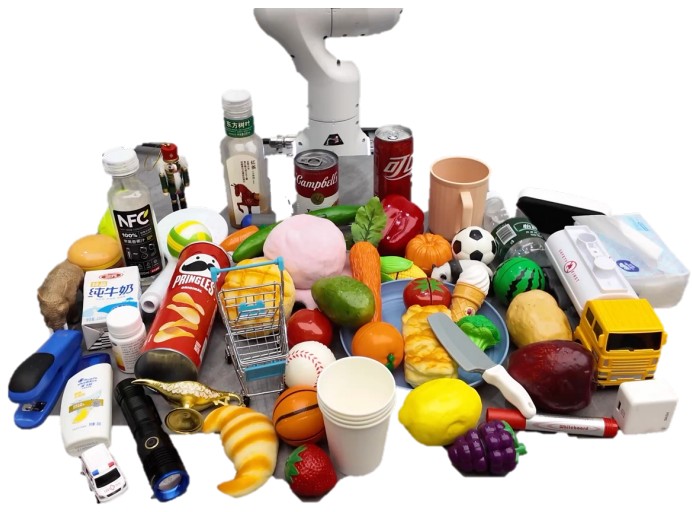

Figure 17: **The real-world assets used in our real-world experiments**. More than 100 diverse objects are used in our 6-DoF rearrangement experiments.

# D    Additional Implementation Details

## D.1    Detail Real World Experiment Results

To fully demonstrate the generalization of SOFAR rather than cherry-picking, we carefully design 60 different real-world experimental tasks, covering more than 100 different and diverse objects. Similar to the Open6DOR [25] benchmark in the simulator, we divide these 60 tasks into three parts: position-track, orientation-track, and the most challenging comprehensive & 6-DoF-track. Each track is further divided into simple and hard levels. The position-simple track includes tasks related to front & back & left & right spatial relationships, while the position-hard track includes tasks related to between, center, and customized. The orientation-simple track includes tasks related to the orientation of object parts, and the orientation-hard track includes tasks related to whether the object is upright or flipped (with very strict requirements for angles in both upright and flipped cases). Comprehensive tasks involve complex instruction understanding and long-horizon tasks; 6-DoF tasks simultaneously include requirements for both object position and orientation instructions. In Table 13, we present the complete task instructions, as well as the performance metrics of SOFAR and the baseline. Due to the large number of tasks, we performed each task three times. It can be seen that SOFAR achieves the best performance in all tracks, especially in the orientation-track and comprehensive & 6-DoF-track. We also show all the objects used in the real-world experiments in Fig. 17, covering a wide range of commonly and uncommonly used objects in daily life.

## D.2    PointSO Model Details

For PointSO, we utilize FPS + KNN to perform patchify and employ a small PointNet [86] as the patch encoder. Subsequently, a standard Transformer encoder is adopted as the backbone, followed by a single linear layer to map the output to a three-dimensional vector space. All parameter configurations follow prior work on point cloud representation learning [26, 89, 91]. De-

Table 15: **Details of PointSO model variants**. This table format follows Dosovitskiy et al. [28].

| Model | CLIP | Layers | Hidden size | MLP size | Heads | #Params |
|-------|------|--------|-------------|----------|-------|---------|
| Small | ViT-B/32 | 12 | 256 | 1024 | 4 | 11.4M |
| Base  | ViT-B/32 | 12 | 384 | 1536 | 6 | 19.0M |
| Large | ViT-B/32 | 12 | 512 | 2048 | 8 | 43.6M |

tailed hyperparameter and model configurations are provided in Tables 14 and 15.

## D.3    SoFar-LLaVA Model Details

In addition to leveraging the extensive knowledge and strong generalization capabilities of closed-source/open-source pretrained VLMs [98, 1, 3] for zero-shot or in-context learning, SOFAR can also enhance the planning performance of open-source models through visual instruction tuning for rapid

Table 13: **Detailed zero-shot real-world 6-DoF rearrangement results**.

| Task | CoPa [44] | ReKep-Auto [47] | SoFar-LLaVA (Ours) | SoFar (Ours) |
|---|---|---|---|---|
| *Positional Object Manipulation* | | | | |
| Move the soccer ball to the right of the bread. | 2/3 | 3/3 | 3/3 | **3/3** |
| Place the doll to the right of the lemon. | 3/3 | 3/3 | 3/3 | **3/3** |
| Put the pliers on the right side of the soccer ball. | 1/3 | 1/3 | 3/3 | **2/3** |
| Move the pen to the right of the doll. | 3/3 | 2/3 | 3/3 | **3/3** |
| Place the carrot on the left of the croissant. | 2/3 | 3/3 | 2/3 | **2/3** |
| Move the avocado to the left of the baseball. | 3/3 | 2/3 | 2/3 | **3/3** |
| Pick the pepper and place it to the left of the charger. | 1/3 | 2/3 | 2/3 | **2/3** |
| Place the baseball on the left side of the mug. | 3/3 | 2/3 | 2/3 | **3/3** |
| Arrange the flower in front of the potato. | 2/3 | 3/3 | 2/3 | **3/3** |
| Put the volleyball in front of the knife. | 3/3 | 3/3 | 3/3 | **3/3** |
| Place the ice cream cone in front of the potato. | 2/3 | 3/3 | 2/3 | **3/3** |
| Move the bitter melon to the front of the forklift. | 2/3 | 1/3 | 2/3 | **2/3** |
| Place the orange at the back of the stapler. | 3/3 | 2/3 | 3/3 | **3/3** |
| Move the panda toy to the back of the shampoo bottle. | 2/3 | 3/3 | 3/3 | **2/3** |
| pick the pumpkin and place it behind the pomegranate. | **3/3** | 2/3 | 1/3 | 2/3 |
| Place the basketball at the back of the board wipe. | 2/3 | 2/3 | 3/3 | **2/3** |
| Put the apple inside the box. | 3/3 | 2/3 | 3/3 | **3/3** |
| Place the waffles on the center of the plate. | 3/3 | 2/3 | 3/3 | **3/3** |
| Move the hamburger into the bowl. | 2/3 | 2/3 | 2/3 | **3/3** |
| Pick the puppet and put it into the basket. | 1/3 | 2/3 | 2/3 | **2/3** |
| Drop the grape into the box. | 2/3 | 3/3 | 3/3 | **2/3** |
| Put the doll between the lemon and the USB. | 2/3 | 2/3 | 2/3 | **3/3** |
| Set the duck toy in the center of the cart, bowl, and camera. | 2/3 | 1/3 | 2/3 | **2/3** |
| Place the strawberry between the Coke bottle and the glue. | 2/3 | 2/3 | 3/3 | **3/3** |
| Put the pen behind the basketball and in front of the vase. | 2/3 | 1/3 | 2/3 | **2/3** |
| Total success rate | 74.7% | 72.0% | 81.3% | **85.3%** |
| *Orientational Object Manipulation* | | | | |
| Turn the yellow head of the toy car to the right. | 2/3 | 2/3 | 1/3 | **2/3** |
| Adjust the knife handle so it points to the right. | 2/3 | 1/3 | 2/3 | **2/3** |
| Rotate the cap of the bottle towards the right. | 2/3 | 2/3 | 2/3 | **2/3** |
| Rotate the tip of the screwdriver to face the right. | 0/3 | 0/3 | 1/3 | **1/3** |
| Rotate the stem of the apple to the right. | 0/3 | 1/3 | 1/3 | **2/3** |
| Turn the front of the toy car to the left. | 0/3 | 0/3 | 2/3 | **2/3** |
| Rotate the cap of the bottle towards the left. | 2/3 | 1/3 | 1/3 | **2/3** |
| Adjust the pear's stem to the right. | 1/3 | 1/3 | 1/3 | **1/3** |
| Turn the mug handle to the right. | 1/3 | 1/3 | 2/3 | **2/3** |
| Rotate the handle of the mug to towards right. | 2/3 | 1/3 | **2/3** | 1/3 |
| Rotate the box so the text side faces forward. | 0/3 | 1/3 | 0/3 | **1/3** |
| Adjust the USB port to point forward. | 0/3 | 0/3 | 1/3 | **1/3** |
| Set the bottle upright. | 0/3 | 1/3 | 0/3 | **1/3** |
| Place the coffee cup in an upright position. | 1/3 | 1/3 | 2/3 | **2/3** |
| Upright the statue of liberty | 0/3 | 0/3 | **1/3** | 0/3 |
| Stand the doll upright. | 0/3 | 1/3 | 0/3 | **1/3** |
| Right the Coke can. | 0/3 | 0/3 | 1/3 | **1/3** |
| Flip the bottle upside down. | 0/3 | 0/3 | 0/3 | **1/3** |
| Turn the coffee cup upside down. | 0/3 | 0/3 | 1/3 | **1/3** |
| Invert the shampoo bottle upside down. | 0/3 | 0/3 | 0/3 | **0/3** |
| Total success rate | 21.7% | 23.3% | 35.0% | **43.3%** |
| *Comprehensive 6-DoF Object Manipulation* | | | | |
| Pull out a tissue. | 3/3 | 3/3 | 2/3 | **3/3** |
| Place the right bottle into the box and arrange it in a 3×3 pattern. | 0/3 | 0/3 | 0/3 | **1/3** |
| Take the tallest box and position it on the right side. | 1/3 | 1/3 | 3/3 | **3/3** |
| Grasp the error bottle and put it on the right side. | 1/3 | 2/3 | 1/3 | **2/3** |
| Take out the green test tube and place it between the two bottles. | 2/3 | 2/3 | 3/3 | **3/3** |
| Pack the objects on the table into the box one by one. | 1/3 | 1/3 | 0/3 | **1/3** |
| Rotate the loopy doll to face the yellow dragon doll | 0/3 | 1/3 | 1/3 | **1/3** |
| Right the fallen wine glass and arrange it neatly in a row. | 0/3 | 0/3 | 0/3 | **0/3** |
| Grasp the handle of the knife and cut the bread. | 0/3 | 0/3 | 0/3 | **1/3** |
| Pick the baseball into the cart and turn the cart to facing right. | 0/3 | 0/3 | 1/3 | **2/3** |
| Place the mug on the left of the ball and the handle turn right. | 0/3 | 0/3 | 1/3 | **1/3** |
| Aim the camera at the toy truck. | 1/3 | 0/3 | 1/3 | **1/3** |
| Rotate the flashlight to illuminate the loopy. | 0/3 | 0/3 | 1/3 | **1/3** |
| Put the pen into the pen container. | 0/3 | 1/3 | 0/3 | **1/3** |
| Pour out chips from the chips cylinder to the plate. | 0/3 | 1/3 | 1/3 | **1/3** |
| Total success rate | 20.0% | 26.7% | 33.3% | **48.9%** |

fine-tuning. The pipeline of the model is illustrated in Fig. 18. A JSON-formatted 6-DoF scene graph, processed through a text tokenizer, along with the image refined by SoM [129], is fed into an LLM (*e.g.*, LLaMA [110, 111]) for supervised fine-tuning [67]. In the Open6DOR [25] task, we supplement the training dataset with additional samples retrieved and manually annotated from Objaverse [20], ensuring alignment with the object categories in the original benchmark. This dataset

Table 14: **Training recipes for PointSO and SOFAR-LLaVA.**

| Config | PointSO | | | | SOFAR-LLaVA |
| | Small | Base | Large | Finetune | SFT |
| --- | --- | --- | --- | --- | --- |
| optimizer | AdamW | AdamW | AdamW | AdamW | AdamW |
| learning rate | 5e-5 | 5e-5 | 2e-5 | 5e-5 | 2e-5 |
| weight decay | 5e-2 | 5e-2 | 5e-2 | 5e-2 | 0 |
| learning rate scheduler | cosine | cosine | cosine | cosine | cosine |
| training epochs | 300 | 300 | 300 | 50 | 2 |
| warmup epochs | 10 | 10 | 10 | 5 | 0.03 |
| batch size | 256 | 256 | 256 | 256 | 128 |
| drop path rate | 0.2 | 0.2 | 0.2 | 0.2 | - |
| number of points | 10000 | 10000 | 10000 | 10000 | - |
| number of point patches | 512 | 512 | 512 | 512 | - |
| point patch size | 32 | 32 | 32 | 32 | - |
| augmentation | Rot&Part&Noise | Rot&Part&Noise | Rot&Part&Noise | Rotation | - |
| GPU device | 8×H800 | 8×H800 | 8×H800 | 8×H800 | 8×H800 |

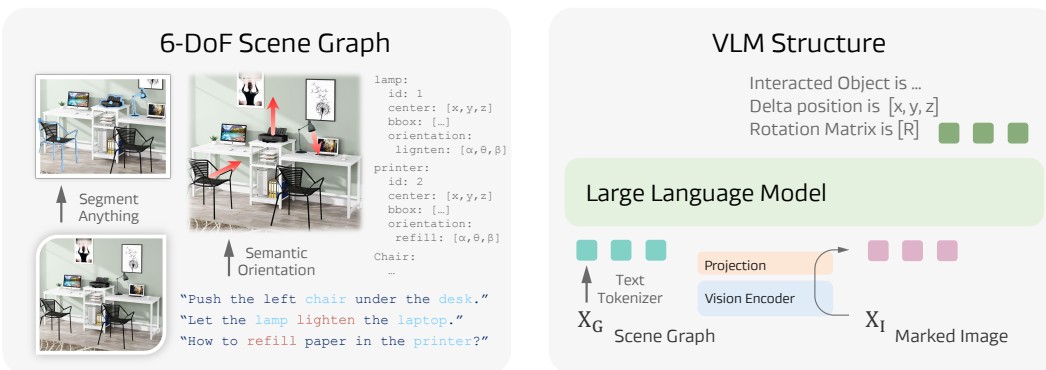

Figure 18: **Pipeline of SOFAR-LLaVA**, a fine-tuned VLM based on visual instruction tuning.

includes approximately 3,000 6-DoF object manipulation instructions. Using this data, we construct dialogue-style training data based on ChatGPT and train the SOFAR-LLaVA model. The training hyperparameters are detailed in Table 14. Similarly, we finetune PointSO on this training dataset and achieve superior performance on the Open6DOR task.

### D.4 ChatGPT API Costs

The knowledge of OrienText300K is derived from the annotations of 3D modelers on Sketchfab, combined with ChatGPT's filtering and comprehension capabilities. To generate semantic orientation annotations, we filter the 800K dataset of Objaverse [20] and apply ChatGPT to approximately 350K of the filtered data to generate semantic text-view index pairs. The OpenAI official API was used for these calls, with the GPT-4o version set to 2024-08-06 and the output format configured as JSON. The total cost for debugging and execution amounted to approximately $10K.

## E Additional Benchmark Statistic Analysis

### E.1 6-DoF SpatialBench Analysis

We conduct a statistical analysis of the manually constructed 6-DoF SpatialBench, with category comparisons and word cloud visualizations shown in Fig. 19. We collect diverse image data from the internet, encompassing scenes such as indoor, outdoor, and natural landscapes. The questions may involve one or multiple objects, with varying levels of uncertainty in image resolution. Most importantly, we are the first to propose a VQA benchmark for orientation understanding, focusing on both quantitative and qualitative evaluation of orientation.

### E.2 Open6DOR V2 Analysis

Open6DOR V2 builds upon Open6DOR V1 by removing some incorrectly labeled data, removing manual evaluation metrics, and integrating assets and metrics into Libero, enabling closed-loop policy evaluation. The detailed number of tasks is presented in Table 16, comprising over 4,500 tasks in total. Notably, we remove level 2 of the position track in Open6DOR V1 [25] because it requires manual inspection, which is not conducive to open-source use and replication by the community. Besides, due to the randomness of object drops in the scene, approximately 8% of the samples already satisfy the evaluation metrics in their initial state.

## F  Related Works

### F.1  Vision-Language Models for Spatial Understanding

Vision-Language Models are rapidly being developed in the research community, driven by the storming lead in extending GPT-style [95, 96, 7] Large Language Models (LLMs) like LLaMA [110, 111] to VLMs [67, 68, 27, 105, 141, 144, 51]. SpatialVLM [10] pioneers this direction by constructing VQA data in spatial understanding from RGB-D, which is used for training an RGB-only VLM. Following SpatialVLM, SpatialRGPT [12] extends RGB-based spatial understanding to RGB-D by constructing spatial understanding data using 3D scene graphs. SpatialBot [8] explores RGB-D spatial reasoning through hierarchical depth-based reasoning. Some other works propose visual prompting for improving GPT-4V's spatial understanding [65, 129, 75]. Meanwhile, another line of works explores VLMs using 3D representations such as point clouds for 3D scene [43, 35] and object-centric [91, 127, 92] understanding. More recently, OmniSpatial [51] proposed a comprehensive and challenging spatial reasoning benchmark. Despite the remarkable progress, these works are limited to 3-DoF understanding, which is not actionable. In contrast, we explore spatial understanding in 6-DoFs from RGB-D via VLMs. Unlike vanilla 3D scene graphs used by SpatialRGPT for data construction, we propose orientation-aware 3D scene graphs realized by our proposed PointSO. In addition, we formulate spatial understanding as graph learning, where the scene graph nodes are directly input during inference.

### F.2  Language-Grounded Robot Manipulation

Language-grounded robot Manipulation adopts the human language as a general instruction interface. Existing works can be categorized into two groups: **i)** *End-to-end* models like RT-series [6, 152, 4] built upon unified cross-modal Transformers with tokenized actions [100, 66, 145], large vision-language-action models built from VLMs [56, 143], or 3D representations [146, 137]. Training on robot data such as Open X-Embodiment [15] and DROID [55], a remarkable process has been made. However, the data *scale* is still limited compared to in-the-wild data for training VLMs. **ii)** *Decoupled* high-level reasoning and low-level actions in large VLMs and small off-the-shelf policy models, primitives [49, 63, 46, 44, 34, 131, 138, 77, 39, 149], or articulated priors [45, 60]. Our SOFAR lies in this group, where an open-world generalization property emerges from VLMs and our proposed PointSO is empowered by orientation-aware spatial understanding.

### F.3  3D Representation Learning

Research on 3D Representation Learning encompasses various methods, including point-based [86, 87], voxel-based [78], and multiview-based approaches [102, 38]. Point-based methods [93, 31] have gained prominence in object classification [122, 113] due to their sparsity yet geometry-informative representation. On the other hand, voxel-based methods [21, 134, 90] offer dense representation and translation invariance, leading to a remarkable performance in object detection [17] and segmentation [133, 2]. The evolution of attention mechanisms [114] has also contributed to the development of effective representations for downstream tasks, as exemplified by the emergence of 3D Transformers [31, 74, 76]. Notably, 3D self-supervised representation learning has garnered significant attention in recent studies. PointContrast [126] utilizes contrastive learning across different views to acquire discriminative 3D scene representations. Innovations such as Point-BERT [136] and Point-MAE [83] introduce masked modeling [40, 22] pretraining into the 3D domain. ACT [26] pioneers cross-modal geometry understanding through 2D or language foundation models such as CLIP [97] or BERT [22]. Following ACT, RECON [89] further proposes a learning paradigm

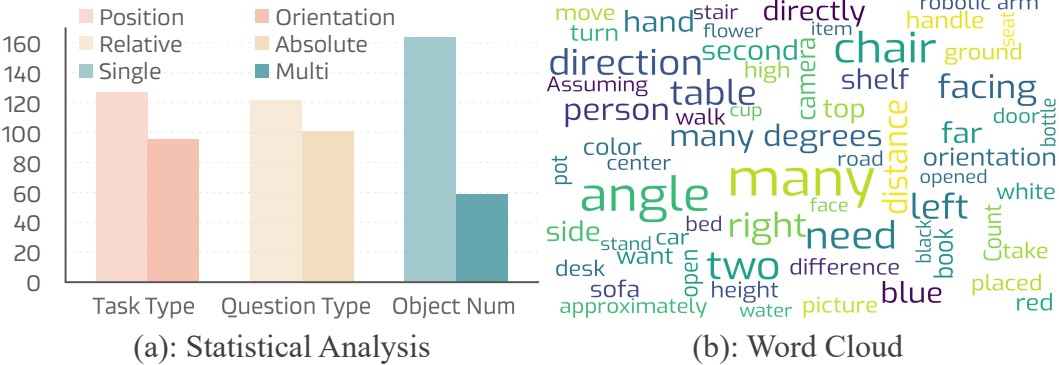

(a): Statistical Analysis    (b): Word Cloud

Figure 19: **6-DoF SpatialBench statistics**. (a) Statistical analysis of the task type, question type, and object relation. (b) Word cloud visualization.

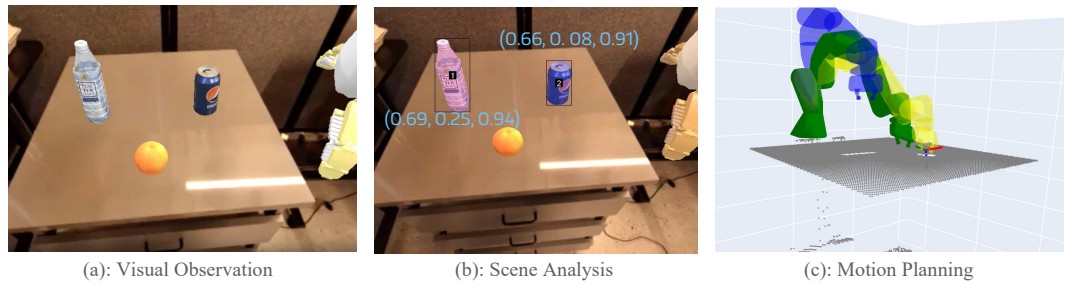

(a): Visual Observation          (b): Scene Analysis          (c): Motion Planning

Figure 20: An example of SOFAR how to finish "move near" task in SIMPLER [62].

that unifies generative and contrastive learning. PPT [142] highlights the significance of positional encoding in 3D representation learning. Additionally, leveraging foundation vision-language models like CLIP [26, 97] has spurred the exploration of a new direction in open-world 3D representation learning. This line of work seeks to extend the applicability and adaptability of 3D representations in diverse and open-world/vocabulary scenarios [84, 140, 24, 23, 32].

## G   Additional Discussions

### G.1   Relation to Affordance & 6-DoF Pose Estimation

Conceptually, this semantic orientation is a counterpart of *affordance* [36, 101, 88, 37] but beyond, as SO and affordance, all present potential actions and interactions with objects. However, SO also contains the spatial understanding of intra-object part-level attributes more than affordance learning. Compared to vanilla 6-DoF pose estimation, our proposed SO combined with the 3-DoF translation understanding, has the same DoF completeness. The difference is, our proposed SO is grounded in languages, making it useful for open-world manipulation requiring complicated spatial reasoning [109, 49, 25]. In addition, our Semantic Orientation can be auto-labeled from Internet 3D data that achieves higher scalability, as introduced in Section 2.

### G.2   Comparison to Concurrent Works

**Comparison with ReKep [47]**

Recently, ReKep has succeeded in executing complex robotic tasks, such as long-horizon manipulation, based on the relationships and constraints between spatial key points. Its structural design offers many insights that SOFAR can draw upon, yet it also presents several issues: (1) Overly customized prompt engineering. ReKep requires manually designed complex system prompts for each task during inference. While this approach may be described as "no training", it cannot be considered a true zero-shot transfer. In contrast, SOFAR achieves genuine zero-shot transfer by eliminating the need for any human involvement during inference; (2) Using constraints based solely on key

Table 16: **Statistics of Open6DOR V2 Benchmark.** The entire benchmark comprises three independent tracks, each featuring diverse tasks with careful annotations. The tasks are divided into different levels based on instruction categories, with statistics demonstrated above.

| Track | Position-track | | | | | | | Rotation-track | | | 6-DoF-track | Totel |
|---|---|---|---|---|---|---|---|---|---|---|---|---|
| Level | Level 0 | | | | | Level 1 | | Level 0 | Level 1 | Level 2 | - | - |
| Task Catog. | Left | Right | Top | Behind | Front | Between | Center | Geometric | Directional | Semantic | - | - |
| Task Stat. | 296 | 266 | 209 | 297 | 278 | 193 | 159 | 318 | 367 | 134 | 1810 | 4535 |
| Benchmark Stat. | 1698 | | | | | | | 1027 | | | 1810 | 4535 |

points fails to capture the full 6-DoF pose integrity of objects. For example, in the "pouring water" task, merely bringing the spout of the kettle close to the cup may lead to incorrect solutions, such as the kettle overturning; (3) ReKep requires all key points to be present in the first frame, and each step of the process—from mask extraction to feature dimensionality reduction, clustering, and filtering—introduces additional hyperparameters.

**Comparison with Orient Anything [118]**

Recently, Orient Anything also highlighted the importance of orientation in spatial perception and adopted a training data construction approach similar to Our PointSO. Our primary distinction lies in semantic orientation, which is language-conditioned orientation. In contrast, Orient Anything is limited to learning basic directions such as "front" and "top". By aligning with textual information, semantic orientation better enhances spatial perception, understanding, and robotic manipulation.

### G.3   Future Works

Future work includes further expanding the OrienText300K with larger datasets like Objaverse-XL [19], enhancing the performance of semantic orientation through self-supervised learning and pretraining methods [40, 97, 26, 89], and demonstrating its effectiveness in a broader range of robotic scenarios, such as navigation [9], mobile manipulation [132], lifelong learning [64], spatio-temporal reasoning [47, 72, 73, 130], humanoid [41, 11, 14, 42], and human-robot interaction [70, 71].

## H   Additional Visualizations

### H.1   Robotic Manipulation

As shown in Fig. 20, we present a visualization of executing a task named "move near". According to the input image and task instruction - "*move blue plastic bottle near pepsi can*", SOFAR can predict the center coordinate of the target object (bottle) and relative target (pepsi can), and it would infer the place coordinate and produce a series of grasp poses.

### H.2   6-DoF SpatialBench

To further evaluate 6-DoF spatial understanding, we construct a 6-DoF SpatialBench. We present examples of question-answer pairs from the 6-DoF SpatialBench, with quantitative and qualitative questions shown in Figs. 21 and 22, respectively. The benchmark we constructed is both challenging and practical, potentially involving calculations based on the laws of motion, such as "*Assuming a moving speed of 0.5 m/s, how many seconds would it take to walk from here to the white flower?*" Moreover, it covers a wide range of spatially relevant scenarios across both indoor and outdoor environments.

### H.3   System Prompts

Prompt engineering significantly enhances ChatGPT's capabilities. The model's understanding and reasoning abilities can be greatly improved by leveraging techniques such as Chain-of-Thought [119] and In-Context Learning [7]. Figs. 23 and 24 illustrate the system prompt we used in constructing OrienText300K. Fig. 25, Fig. 26, and Fig. 27 illustrate the system prompt we used when evaluating SOFAR on Open6DOR (simulation), object manipulation (both simulation and real worlds), and

VQA, respectively. Note that different from previous methods [46, 47], SoFAR does not require complicated in-context examples.

# I  Broader impacts

Our work on semantic orientation significantly enhances robotic spatial reasoning and manipulation capabilities, enabling more intuitive human-robot interaction. This advancement can improve efficiency in various industries, such as manufacturing and healthcare, and enhance the quality of life by assisting in tasks like elderly care and home automation. Additionally, it contributes to the broader field of AI research by providing new tools and benchmarks for spatial reasoning and language-grounded manipulation.

**[Task Type: Position     Question Type: Absolute]**

[**Question**]: Count from right to left and start at 1, which two of the red flower pots are the group of people in the middle of?

[**A**]: "4 and 5"
[**B**]: "2 and 3"
[**C**]: "1 and 2"
[**D**]: "3 and 4"

[**Answer**]: C

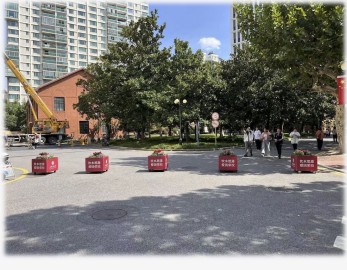

**[Task Type: Orientation     Question Type: Absolute]**

[**Question**]: If you want to align the orientations of the two chairs, what is the minimum angle you need to rotate the chair on the right?

[**A**]: "75°"
[**B**]: "55°"
[**C**]: "35°"
[**D**]: "15°"

[**Answer**]: C

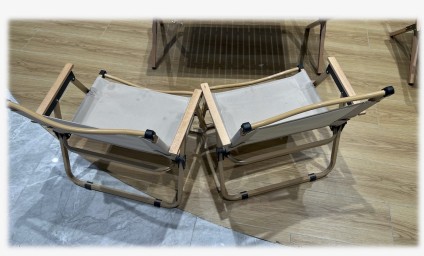

**[Task Type: Position     Question Type: Absolute]**

[**Question**]: Assuming a moving speed of 0.5 m/s, how many seconds would it take to walk from here to the white flower?

[**A**]: "3s"
[**B**]: "5s"
[**C**]: "7s"
[**D**]: "10s"

[**Answer**]: B

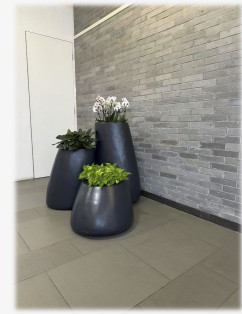

**[Task Type: Orientation     Question Type: Absolute]**

[**Question**]: How many white chairs are facing the window?

[**A**]: "2"
[**B**]: "4"
[**C**]: "1"
[**D**]: "3"

[**Answer**]: A

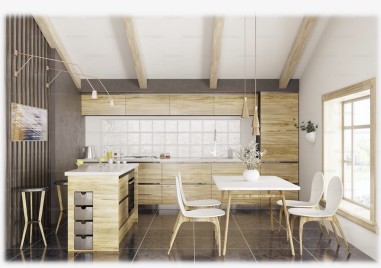

Figure 21: **Visualization example of 6-DoF SpatialBench data samples.**

**[Task Type: Position    Question Type: Relative]**

[**Question**]: Which side of the steps is narrower?

**[A]:** "the left"
**[B]:** "the right"
**[C]:** "the middle"
**[D]:** "the same"

**[Answer]:** B

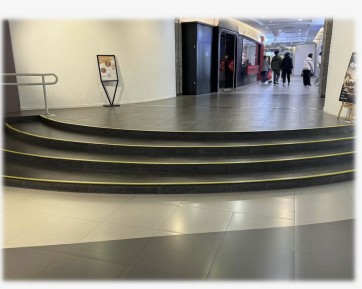

**[Task Type: Orientation    Question Type: Relative]**

[**Question**]: Which direction does the handle of the cup in the upper right corner point to?

**[A]:** "left"
**[B]:** "right"
**[C]:** "front"
**[D]:** "back"

**[Answer]:** A

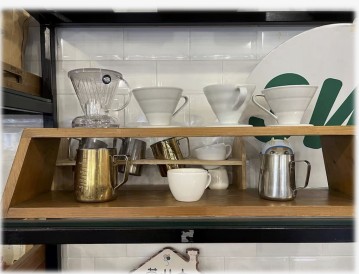

**[Task Type: Position    Question Type: Relative]**

[**Question**]: How many compartments are there in the heart-shaped grid for storing books?

**[A]:** "5"
**[B]:** "3"
**[C]:** "6"
**[D]:** "4"

**[Answer]:** A

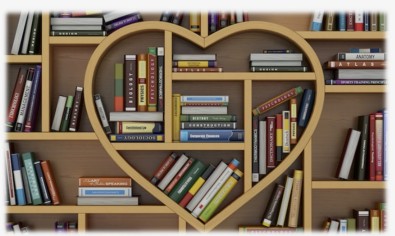

**[Task Type: Orientation    Question Type: Relative]**

[**Question**]: If you are a driver driving a car on the road from near to far, which direction will you turn to?

**[A]:** "first turn left and then left"
**[B]:** "first turn right and then left"
**[C]:** "first turn left and then right"
**[D]:** "first turn right and then right"

**[Answer]:** C

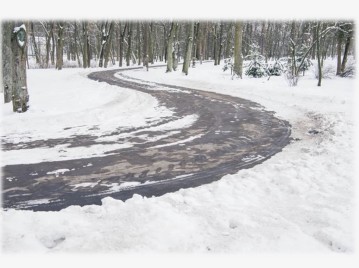

Figure 22: **Visualization example of 6-DoF SpatialBench data samples.**

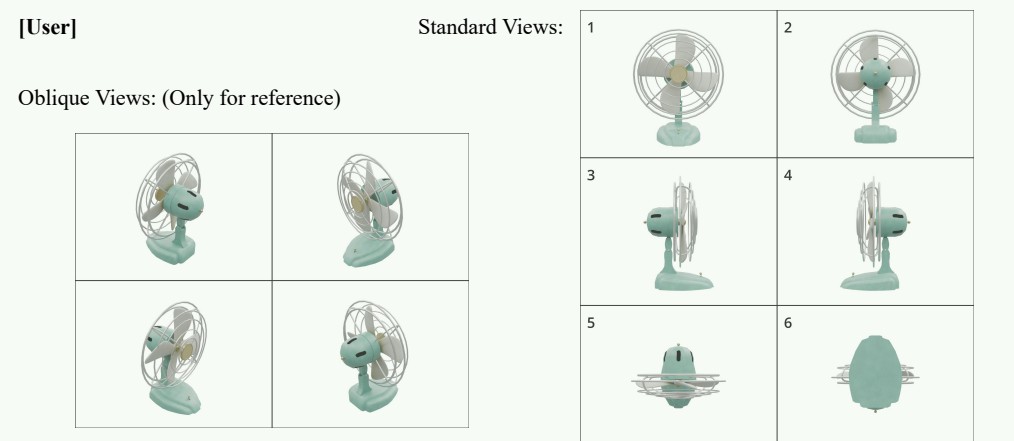

**[System Prompt]**

You are an expert AI assistant for 3D object understanding.
The user imported a potentially uncalibrated 3D model into Blender and placed cameras in front, back, left, right, top, and bottom to render images, labeled from 1 to 6.
You are required to infer the entire 3D object based on these images and determine its attributes.

Your task is to assess the following attributes for each 3D model and respond with "true" or "false" for each question:
Axis Alignment: Determine whether the object is horizontally and vertically aligned across all views. Key features (e.g., edges, handles, or other distinct elements) of the object must be perpendicular or parallel to the cameras. Respond "true" if all views are aligned with the axis, "false" if not.
Scene or Collection: Determine whether the 3D model represents a 3D scene or a collection of independent objects (e.g. a room, outdoor scene, or multiple independent objects). Respond with "true" if it does, and "false" if it only contains a single object.
White: Determine whether the 3D model only has single white or gray colors, and lacks any other colors. Respond with "true" if it is white or gray, and "false" if it has any other colors (e.g., black or yellow).
Ground: Determine whether the 3D model includes a ground plane for auxiliary visualization. Respond with "true" if it does, and "false" if it only has the object.
High Quality: Determine whether the 3D model is a refined, well-constructed mesh without defects, such as point noise or streaking artifacts commonly found in low-quality RGBD scans. Respond with "true" if the mesh is clean and smooth, and "false" if it contains noise, roughness, or visual artifacts.
Distinguishable Views: Determine whether the 3D model has distinguishable views, or has clear semantic information in certain views (e.g., some 3D object has clear front, top directions). Respond with "true" if the 6 views show noticeable differences or have clear semantic information in certain views, and "false" if the views appear identical and there is no obvious semantic information on all views.
Reasonable Object: Determine whether the 3D model represents a common, recognizable, meaningful object. Respond with "true" if it is, and "false" if it is abstract, confused, or unrecognizable.

You need to first analyze the 3D object detail, and then output its correct attributes.

Figure 23: **The system prompt of GPT-4o used for filtering Objaverse data.**

**[System Prompt]**

You are a visual assistant specializing in interpreting 3D objects from multiple perspectives.
You will receive 6 images of a 3D object from standard views (front, back, left, right, top, bottom), presented in random order. Typically, image 5 corresponds to the top view.
Your task is to generate an instruction-index pair that identifies a meaningful semantic orientation for the object, based on its function or commonly understood orientation.
The instruction can be a verb, noun, adjective, or phrase, and must clearly relate to the object's function or orientation in everyday use.
Ensure the direction is clear, objective, and uniquely meaningful.

Examples:
For a pen, the instruction might be "pen cap", and the index is the image with the pen cap facing the camera.
For a cup, the instruction might be "handle", and the index is the image with the cup handle facing the camera.
For a phone, the instruction might be "screen", and the index is the image with the phone screen facing the camera.
For a table, the instruction might be "on", and the index is the image with the tabletop facing the camera.
For a power outlet, the instruction might be "plug-in". Based on common knowledge, its semantic orientation is perpendicular to the power outlet's plane, along the direction of the power outlet's slots, and therefore, the index is the image with the power outlet pinholes' plane facing the camera.
For a desk, the instruction might be "open the drawer". Based on common sense, the robot would need to pull the drawer open. The semantic orientation corresponds to the direction of the drawer's extension, hence the index is the image with the drawer handle facing the camera.
For a microphone, the instruction might be "speak", the semantic orientation is along the direction of the microphone's head, and therefore, the index is the image with the microphone head facing the camera.

You need to first analyze the category, attributes, characteristics, state, and usage of this 3D object in detail, and then output a pair of instructions and index.
When it is challenging to generate complex instructions, or when multiple views of the object are too similar to produce a unique instruction, you can use simpler instructions, such as "top" or "front".
The output format is as follows:
Analysis: "..."
Instruction: "..."
Index: 1-6

Standard Views:

**[User]**

Oblique Views: (Only for reference)

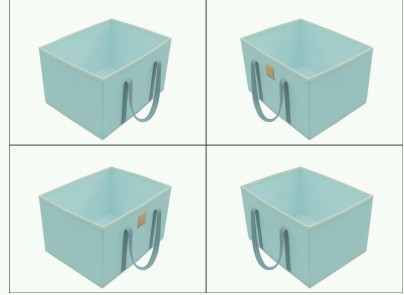

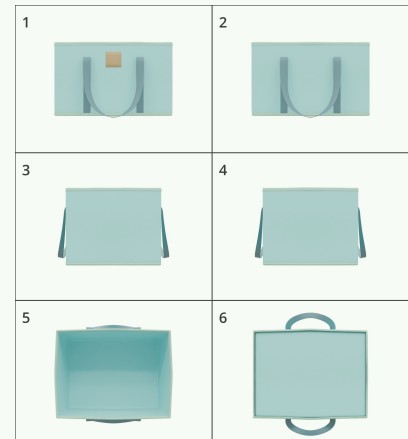

Figure 24: **The system prompt of GPT-4o used for generating semantic orientation-Index pairs.**

**[Parsing System Prompt]**

You are an assistant specialized in interpreting tabletop pick-and-place instructions for robotic manipulation.
Your main goals are to identify relevant objects and analyze necessary orientations.

Key Objectives
1. Object Identification: Identify and list the objects mentioned in the instruction. Exclude the table itself.
2. Orientation Analysis: For the object needs to pick & place, determine any required orientation crucial to the task's success. If orientation isn't specified, leave the orientation list empty.
3. Direction Terms: Limit directional terms to these two categories:
  - Object Parts: e.g., "handle", "pen cap", "top"
  - Interaction Actions: e.g., "pour out", "open"
  Terms must be single words, not phrases or sentences.
  You must analysis both the instruction and the image to determine the object's direction attributes.
4. Disambiguation of Identification: If instructions reference vague objects (e.g., "else object", "all objects"), use visual information to clarify.
5. Disambiguation of Orientation: If the instructions describe complex rotation like "upright", you can interpret them as ensuring an object's relevant part is aligned with the z-axis (e.g., "bottle cap", "top").
  This disambiguation utilizes world knowledge, as we define the far-to-near direction as the x-axis, the left-to-right direction as the y-axis, and the bottom-to-top direction as the z-axis.
  Similarly, place an object to point forward means that the "top" of the object is oriented along the x-axis.

**[Reasoning System Prompt]**

You are an assistant for spatial intelligence and robotic operations, specializing in pick-and-place tasks.
Your role is to process robotic commands to pick a object and place it in a specific location.

Input Context:
1. Pick & Place Command: A directive specifying which object to pick and where to place it, including any specific pose requirements.
2. picked object info: A dictionary with the picked object's position in the world coordinate system.
  - Coordinates: Object center and bounding box in 3D (x, y, z), where:
    -- x: Extends from far to near. Objects closer to the observer have larger x-values
    -- y: Extends from left to right. Objects further to the right have larger y-values
    -- z: Extends upward. Objects positioned higher have larger z-values
3. other objects info: A list of dictionaries with the position of other objects in the scene.
  - Coordinates: Object center and bounding box in 3D (x, y, z), same in the world coordinate system.

Objective:
1. Generate target placement position: Based on the spatial location descriptions provided in the instructions (e.g., 'behind,' 'between,' 'left,' etc.), as well as each object's center and bounding box (bbox), analyze and calculate the appropriate placement for the picked object.
  - front indicates positioning the object at an x-coordinate slightly larger than the reference object's x maximum.
  - right indicates positioning the object at a y-coordinate slightly larger than the reference object's y maximum.
  - between indicates positioning the object at the midpoint between two reference objects.

**[User]**

Place the knife behind the clipboard on the table.
And rotate the handle of the knife to left.

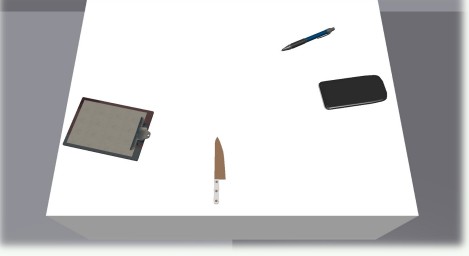

Figure 25: **The system prompt of Open6DOR tasks.**

**[Parsing System Prompt]**

You are a spatially intelligent AI specializing in interpreting objects, spatial directions, and interaction semantics for tasks involving spatial understanding or robotic manipulation.
The user will input an image and an instruction. Analyze user instruction and provide:

Objects: List involved objects using concise nouns or phrases, without any adjectives (e.g, the "top drawer" should be listed as "drawer").
Semantic Orientations: Identify essential spatial or action-related terms, categorized as:
- Object Parts: e.g., "handle", "lid", "top".
- Action Terms: e.g., "pour out", "open".

Guidelines:
Focus on key spatial or action contexts for task completion.
Use implicit spatial conventions (Certain user instructions need to satisfy implicit constraints related to position and orientation.) if practical.
Avoid numeric values or absolute positions.
Only specify object-centric pose relationships, not inter-object positions (such as left, right, front, behind).

**[Reasoning System Prompt]**

You are a robotic spatial intelligence and manipulation assistant, specialized in interpreting commands and scene structures for robotic object manipulation.
Your task is to analyze the user's directive and scene graph to guide the robot in identifying objects, computing spatial transformations, and producing step-by-step guidance for manipulation tasks.

Input Context:
1. Manipulation Command: A directive specifying which object to pick and where to place it, including any specific pose requirements.
2. Scene Graph: A dictionary with the scene objects' position and orientation in the world coordinate system.
   - Coordinates: Object center and bounding box in 3D (x, y, z), where:
     -- x: Extends from near to far. Objects further to the observer have larger x-values
     -- y: Extends from right to left. Objects further to the left have larger y-values
     -- z: Extends upward. Objects positioned higher have larger z-values
   - Orientations of the object's parts (e.g., 'screen', 'handle') in 3D space.
     -- (1, 0, 0): Points forward along the x-axis
     -- (0, 1, 0): Points left along the y-axis
     -- (0, 0, 1): Points upward along the z-axis

Objective: To process each command, follow these steps:

Target Identification: Identify the object to be picked up or manipulated.
Final Position: Specify the intended final position of the object after manipulation, in terms of x, y, z coordinates.
Orientation Mapping: For each semantic orientation provided, compute the final orientation of the manipulated object in the world coordinate system.

**[User]**

Open top drawer.

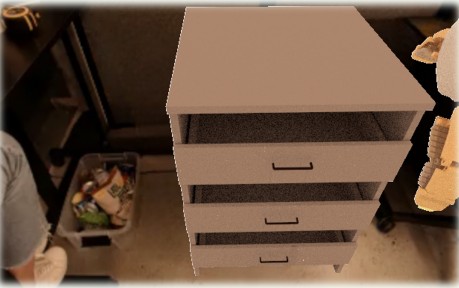

Figure 26: **The system prompt of general manipulation tasks.**

**[Parsing System Prompt]**

You are a spatially intelligent, embodied AI brain specialized in spatial and interactive understanding, tasked with interpreting objects, spatial directions, and relevant interaction semantics in response to the user's queries. The user provides commands or questions related to spatial intelligence or robotic manipulation, often with an image input. Your job is to analyze the given instruction and provide a list of objects involved in the task, alongside semantic orientations needed to complete the instruction effectively. You should focus on the key interaction directions required for successful completion without specifying numeric values or absolute positions, as these will be calculated by an expert model later.

Guidelines:
1. Focus on Semantic Orientations: Define directions concisely using single terms that fall into one of these two categories:
    - Object Parts (e.g., "handle", "screen", "top")
    - Action-Oriented Terms (e.g., "pour out", "plug-in", "open")
2. Optimize for Simplicity: Choose terms that provide essential spatial or action context while remaining simple and intuitive for the model. Use only the most relevant directions or parts needed to complete the user's task.
3. Analysis: When necessary, use implicit spatial conventions where appropriate to ensure a practical output for the model.
4. Only object-centric pose related: Distinguish which object relationships are determined by position (such as left, right, front, behind) and which are determined by object pose, and we only focus on the direction of object - centric pose.

**[Reasoning System Prompt]**

You are a spatial intelligence assistant specialized in understanding 3D visual scenes and answering spatial reasoning questions.

The user will provide:
Image: An image of the scene.
Question: User question about the spatial relationships between objects in the scene.
Scene Graph: Additional information about the objects, including:
    - id: object ID
    - object name: object category
    - center: 3D coordinates of the object's center
    - bounding box: 3D bounding box coordinates
    - orientation: object directions in 3D space

All the coordinates are in the camera coordinate system, where:
    - x-axis: Extends from left to right in the image, objects located right have larger x-values
    - y-axis: Extends from bottom to top in the image, objects located at top of the image have larger y-values
    - z-axis: Extends from near to far in the image, objects located further away have larger z-values

You need to focus mainly on the image, the scene graph information is just for reference.
Avoid providing answers such as "cannot determine." Instead, provide the most likely answer based on the information available.

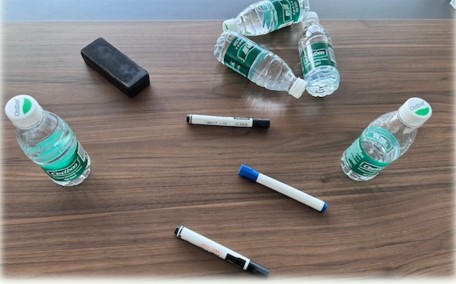

**[User]**

How far between the left bottle and the right bottle?

Figure 27: **The system prompt of visual-question-answering tasks.**

