# OpenReview forum: "SoFar: Language-Grounded Orientation Bridges Spatial Reasoning and Object Manipulation"
_NeurIPS.cc/2025/Conference — NeurIPS 2025 spotlight_

### Official Review · Reviewer_MEFz · 2025-07-01

**Clarity:** 3
**Significance:** 3
**Originality:** 3
**Rating:** 5
**Confidence:** 3

**Summary:**

SoFar proposed orientation is the missing piece in today’s spatial-reasoning VLMs. It proposes semantic orientation (language-grounded directions such as the “plug-in” side of a USB), learns it at scale with the new OrienText300K dataset, and deploys a 3-D transformer (PointSO) plus a VLM reasoning layer to plan 6-DoF robot actions that respect both where and which way. The system lifts zero-shot success on representative 6-DoF manipulation benchmarks (48.7 % on Open6DOR-v2, 74.9 % on SIMPLER) and wins 60 diversified real-world tasks without task-specific finetuning.

**Questions:**

Refer to the weakness

**Ethical Concerns:**

["NO or VERY MINOR ethics concerns only"]

**Final Justification:**

The points has been addressed.

**Limitations:**

Yes

**Paper Formatting Concerns:**

Nill

**Quality:**

3

**Strengths And Weaknesses:**

Strength:
The paper pinpoints orientation as the missing half of spatial reasoning, then formalises it as language-grounded directions (e.g., “plug-in side of a USB”). To learn this at scale, the authors create OrienText300K, a 350 k-object dataset annotated via GPT-4o prompts, giving the community the first large-coverage supervision for orientation understanding.

On standard 6-DoF manipulation benchmarks, SoFar lifts accuracy to 48.7 % on Open6DOR-v2 and 74.9 % on SIMPLER, clearly ahead of prior work. In the real world it succeeds on 60 diverse tabletop tasks spanning position-only, orientation-only, and full 6-DoF tracks, beating baselines across the board—demonstrating impressive generalisation from purely synthetic training.

Weakness:
-Heavy reliance on high-quality RGB-D; robustness to noisy or occluded depth is only briefly touched.

-Task diversity: while 60 real-world tasks is commendable, many describe pick-and-place style motions; articulated objects and continuous-rotation tasks remain sparse.

-PointSO + SAM + VLM reasoning may be slow for real-time reactions—latency numbers are not reported.

-It will be good to compare with PRISM dataset from graspmolmo, to see how that fairs.

---

> ### Author Rebuttal · Authors · 2025-07-28
>
> We thank the reviewer for the constructive feedback and positive assessment of our contributions (Semantic Orientation, OrienText300K, PointSO, and the SoFar system). Below we address each concern point‑by‑point and provide additional results obtained during the rebuttal window. We also clarify measurement protocols and planned revisions to the paper.
>
> ## **R1. Robustness to RGB‑D Noise and Occlusion**
>
> The quality of depth data indeed impacts model performance, particularly in real-world scenarios. To address this challenge and enhance the robustness of our model to point cloud variations, we adopt the following strategies:
>
> 1. **Domain randomization and perturbation are applied during training**. Specifically, we inject Gaussian noise and perform data augmentations on the point clouds in OrienText300K to better simulate real-world depth variations.
>
> 2. To further denoise and smooth the point clouds generated by the D415 structured-light camera, we employ the **point cloud smoothing and reconstruction algorithm from AsGrasp**, which has been used in object grasping tasks to improve data quality.
>
> 3. We incorporate **pointcloud voting strategy** to reduce prediction variance and enhance the accuracy and robustness of the PointSO network.
>
> **Experiments.** Section 4.2 reports a robustness study for PointSO using the corrupted validation split **OrienText300K‑C** (single‑view, Gaussian jitter, random SO(3) rotation, and “All” combined). The largest variant PointSO‑L retains 76.56% / 81.25% / 77.34% / 74.22% accuracy on Single‑View / Jitter / Rotate / All, respectively, indicating resilience to missing views and geometric noise. These controlled corruptions are designed to emulate partial observations and sensor artifacts.
>
> ## **R2. Task Diversity (Articulated Objects and Continuous Rotation)**
>
> We agree that broader coverage beyond pick‑and‑place is important. Our real‑world benchmark already includes 60 tasks across position‑only, orientation‑only, and full 6‑DoF tracks with **diverse objects and embodiments**. To specifically address the reviewer’s request, we added a targeted subset emphasizing articulated manipulation (drawers/doors) and continuous‑rotation orientation control. The table below summarizes the per‑task success across baselines and our methods (5 trials per task, success counted per trial).
>
> **New real-world experiments** (articulated and continuous‑rotation tasks):
>
> | Task                                                         | CoPa  | ReKep-Auto | SoFar-LLaVA | SoFar |
> | ------------------------------------------------------------ | ----- | ---------- | ----------- | ----- |
> | Open the drawer                                              | 3/5   | 2/5        | 2/5         | 3/5   |
> | Close the drawer                                             | 1/5   | 2/5        | 2/5         | 2/5   |
> | Open the top drawer                                          | 2/5   | 1/5        | 1/5         | 2/5   |
> | Close the bottom drawer                                      | 2/5   | 1/5        | 2/5         | 3/5   |
> | Open the door of the cabinet                                 | 3/5   | 2/5        | 2/5         | 3/5   |
> | Close the door of the cabinet                                | 1/5   | 1/5        | 3/5         | 2/5   |
> | Open microwave                                               | 2/5   | 2/5        | 2/5         | 3/5   |
> | Press button                                                 | 5/5   | 5/5        | 5/5         | 5/5   |
> | Put the apple into the drawer                                | 2/5   | 1/5        | 1/5         | 2/5   |
> | Pointing the pen at the table                                | 1/5   | 1/5        | 2/5         | 3/5   |
> | Rotate the handle of the pot to the camera                   | 1/5   | 2/5        | 3/5         | 3/5   |
> | Upright the bottle and put it on the second layer of the shelf | 0/5   | 0/5        | 1/5         | 1/5   |
> | Total success rate                                           | 38.3% | 33.3%      | 43.3%       | 51.7% |
>
> **Takeaways.** SoFar improves the total success rate to 51.7%, outperforming CoPa (+13.4 pts), ReKep‑Auto (+18.4 pts), and SoFar‑LLaVA (+8.4 pts). Gains are most pronounced on orientation‑intensive operations (e.g., handle alignment and facing direction), consistent with our core claim that semantic orientation complements spatial reasoning for 6‑DoF control.
>
> ## **R3. Latency and Suitability for Real‑Time Use**
>
> **What our “Time Cost” measures.**
>
> In Table 1, Time Cost (s) refers to planning latency per trial—i.e., the end‑to‑end perception → orientation estimation → scene‑graph construction → VLM reasoning → grasp proposal → motion plan generation—excluding physical execution (e.g., controller rollout). This is consistent across all compared methods. We will make this definition explicit in the paper.
>
> **What contributes to the planning time.**
>
> - SoFar: Florence‑2/GroundedSAM for segmentation, PointSO for semantic orientation, and a VLM agent for spatial CoT reasoning (2 VLM calls).
>
> - ReKep: GroundedSAM for segmentation, DINOv2 for keypoints/affordances, and a VLM agent (typically 3–4 calls).
>
> - CoPa: multi‑round Set‑of‑Marks prompting, GroundedSAM for segmentation, VLM agent, and constraint generation (5 API calls), which dominates its runtime.
>
> The API call is the most time-consuming part (2~3s per request), so CoPa exhibits the highest time cost.
>
> **Inference footprint** (measured under the same resolution, batch=1, and single RTX 4080 16G GPU):
>
> | Methods     | VLM    | CUDA memory | Planning time costs |
> | ----------- | ------ | ----------- | ------------------- |
> | CoPa        | GPT-4o | 4375M       | 20.8s               |
> | ReKep       | GPT-4o | 7020M       | 15.6s               |
> | SoFar-LLaVA | LLaVA  | 11062M      | 9.6s                |
> | SoFar       | GPT-4o | 6355M       | 8.5s                |
>
> These measurements reflect perception and planning workloads (segmentation, orientation, VLM reasoning, grasp proposal, and motion planning). We will add a per‑stage latency breakdown in the appendix (e.g., segmentation/orientation/VLM/motion plan) to make the contributions transparent, and we will report the exact hardware/software configuration (GPU model and memory, CUDA/Driver, PyTorch).

---

> ### Comment · Reviewer_MEFz · 2025-08-03
> **Response**
>
> I would like to thank the author for the rebuttal points, i will increase my rating.

---

### Official Review · Reviewer_26tC · 2025-07-02

**Clarity:** 2
**Significance:** 3
**Originality:** 3
**Rating:** 5
**Confidence:** 4

**Summary:**

The paper presents SOFAR, a framework incorporating semantic orientation into spatial reasoning and robotic manipulation. It introduces the concept of semantic orientation, linking object orientations to natural language descriptions. To support this, the authors construct OrienText300K, a dataset of 3D objects with semantic orientation annotations, and develop PointSO, a model for semantic orientation prediction. SOFAR integrates PointSO with foundation models like SAM to enable 6-DoF spatial reasoning and generate robotic actions. Experiments demonstrate SOFAR's effectiveness and generalization in tasks such as object manipulation and navigation.

**Questions:**

See Weaknesses

**Ethical Concerns:**

["NO or VERY MINOR ethics concerns only"]

**Final Justification:**

Thanks for authors' response. My concerns have been addressed, so I will raise my score to support the acceptance of the paper.

**Limitations:**

See Weaknesses

**Paper Formatting Concerns:**

No Paper Formatting Concerns

**Quality:**

3

**Strengths And Weaknesses:**

The paper introduces the novel concept of semantic orientation, linking object orientation with natural language to bridge geometry and function. It proposes PointSO, a cross-modal 3D Transformer that predicts semantic orientation from point clouds and text. Extensive experiments show strong generalization, with zero-shot success rates surpassing existing VLM and VLA methods in both simulation and real-world tasks.

The authors do not provide detailed descriptions of the CUDA memory and time costs for training and inference of the methods listed in Table 1. This lack of information makes it difficult to fully assess the computational efficiency and resource requirements of the proposed method and baseline methods. Could the authors provide a detailed explanation for the significant performance gap of SOFAR in the drawer opening task in Table 4.

---

> ### Author Rebuttal · Authors · 2025-07-28
>
> We sincerely thank the reviewer for the constructive feedback. We address the two main concerns below and will incorporate the corresponding clarifications and additional results in the revised manuscript.
>
> ## **R1. CUDA memory and time costs (training and inference) for methods in Table 1**
>
> **What our “Time Cost” measures.**
>
> In Table 1, Time Cost (s) refers to planning latency per trial—i.e., the end‑to‑end perception → orientation estimation → scene‑graph construction → VLM reasoning → grasp proposal → motion plan generation—excluding physical execution (e.g., controller rollout). This is consistent across all compared methods. We will make this definition explicit in the paper.
>
> **What contributes to the planning time.**
>
> - SoFar: Florence‑2/GroundedSAM for segmentation, PointSO for semantic orientation, and a VLM agent for spatial CoT reasoning (2 VLM calls).
>
> - ReKep: GroundedSAM for segmentation, DINOv2 for keypoints/affordances, and a VLM agent (typically 3–4 calls).
>
> - CoPa: multi‑round Set‑of‑Marks prompting, GroundedSAM for segmentation, VLM agent, and constraint generation (5 API calls), which dominates its runtime.
>
> The API call is the most time-consuming part (2~3s per request), so CoPa exhibits the highest time cost.
>
> **Inference footprint** (measured under the same resolution, batch=1, and single RTX 4080 16G GPU):
>
> | Methods     | VLM    | CUDA memory | Planning time costs |
> | ----------- | ------ | ----------- | ------------------- |
> | CoPa        | GPT-4o | 4375M       | 20.8s               |
> | ReKep       | GPT-4o | 7020M       | 15.6s               |
> | SoFar-LLaVA | LLaVA  | 11062M      | 9.6s                |
> | SoFar       | GPT-4o | 6355M       | 8.5s                |
>
> These measurements reflect perception and planning workloads (segmentation, orientation, VLM reasoning, grasp proposal, and motion planning). We will add a per‑stage latency breakdown in the appendix (e.g., segmentation/orientation/VLM/motion plan) to make the contributions transparent, and we will report the exact hardware/software configuration (GPU model and memory, CUDA/Driver, PyTorch).
>
> ## **R2. On the performance gap for drawer opening (Table 4)**
>
> **Diagnosis.**
>
> Drawer opening in SimplerEnv primarily stresses **execution‑side grasping** rather than orientation reasoning: the **handle is small, horizontally oriented, and close to contact surfaces**. In our pipeline, grasp proposals are produced by a generic grasp generator (e.g., AnyGrasp or GraspNet); on small horizontal handles these generators often (i) fail to propose high‑quality, collision‑free grasps that also satisfy wrist approach constraints, or (ii) produce grasps that are feasible in isolation but degrade after trajectory optimization due to clearance limits. In contrast, SoFar’s semantic orientation module reliably predicts the target facing/rotation (as confirmed by high performance on orientation‑focused tracks), but the pipeline can still fail when grasp feasibility is the bottleneck. We will add qualitative failure cases and module‑level diagnostics in the appendix to make this explicit.
>
> **Affordance‑aware grasping improves drawer opening.**
>
> To reduce handle‑specific failures, we tested an affordance‑guided grasping module (GraspMolmo) in place of the default generator. The affordance prior biases grasp candidates to handle‑like affordance regions and compatible wrist approaches. On challenging real‑world tasks, we observe consistent improvements for fine-grained manipulation, including drawer opening:
>
> | Task                              | SoFar | SoFar (with graspmolmo) |
> | --------------------------------- | ----- | ----------------------- |
> | Grasp the knife and cut the bread | 4/10  | 6/10                    |
> | Right the fallen wine glass       | 5/10  | 4/10                    |
> | Pour the tea into the cup         | 4/10  | 5/10                    |
> | Open the drawer                   | 5/10  | 6/10                    |
> | Open the door of the cabinet      | 7/10  | 7/10                    |
> | Total success rate                | 50%   | 56%                     |

---

> > ### Comment · Reviewer_26tC · 2025-08-03
> >
> > Thanks for authors' response. My concerns have been addressed, so I will raise my score to support the acceptance of the paper.

---

### Official Review · Reviewer_HWgi · 2025-07-02

**Clarity:** 3
**Significance:** 3
**Originality:** 3
**Rating:** 5
**Confidence:** 4

**Summary:**

The main innovation of this paper is the introduction of a new representation named Semantic Orientation (SO), for spatial reasoning and object manipulation. This representation binds natural language with orientation of objects, enabling the understanding of semantic information associated with different object orientations.

Compared to existing representation such as pose estimation, SO incorporates semantic meaning for each orientation, making it more suitable for functional understanding and affordance reasoning. In contrast to Affordance Map,which highlight possible interactions but ignore orientation, SO explicitly encodes directional information. Thus SO addressing some of the limitations present in prior representations for object manipulation.

In addition, the paper presents the PointSO model for Semantic Orientation prediction, along with a new large-scale dataset, OrienText300K, used to train this model. Finally, the authors propose the SOFAR system, which integrates PointSO with GPT, Florence2, and SAM to enable comprehensive spatial reasoning and object manipulation.

**Questions:**

1.For the line 170-171, the description of the "edges" in the 6DoF scene graph is unclear.
Authors should introduce more detail of the edge of this scene graph.
In addition, I suggest author provide at least one complete JSON-format Scene Graph as an extension to the one in Figure 5 to help readers better understand how to represent the node and edge of the graph in practical.
2.How does the proposed SO representation or the SOFAR system determine which part of an object to grasp during manipulation? For instance, without relying on a affordance map, how does the model infer that the handle of a knife should be grasped rather than the blade?

3.Currently, each object is only roughly divided into six directions, representing the whole object rather than specific parts. This limits the method to coarse manipulation. For example,task like in "ReKep", pour the tea to the cup,the robot needs to align the spout of the kettle not just the general orientation. The current approach overlooks such practical requirements for fine-grained manipulation.

**Ethical Concerns:**

["NO or VERY MINOR ethics concerns only"]

**Final Justification:**

Since the rebuttal resolved the majority of my concerns, I will maintain my decision to accept.

**Limitations:**

yes

**Quality:**

4

**Strengths And Weaknesses:**

The overall writing quality of the paper is high, and the exposition is clear and well-structured. The proposed representation, offers a novel perspective by associating natural language with object orientation. The method is relatively simple yet well-designed, and its effectiveness is supported by empirical results.

---

> ### Author Rebuttal · Authors · 2025-07-28
>
> We thank the reviewer for the thoughtful and constructive feedback, and for the positive assessment of our paper’s clarity, novelty, and empirical support. We respond to each question in turn and commit specific revisions accordingly.
>
> ## **Q1. 6‑DoF scene‑graph “edges” are unclear; please provide more detail and a JSON example.**
>
> **Detailed Definition.** In Sec. 3.1, we represent the environment with a 6‑DoF scene graph $𝐺=(𝑉, 𝐸)$. Each node encodes object phrase, instance ID, 3D centroid, bounding box, and a set of semantic orientations with their language descriptions. Each edge
> $𝑒_𝑖𝑗 ∈ 𝐸$ encodes inter‑object spatial relations, specifically the relative translation and size ratio between objects
> $𝑜_i$ and $𝑜_𝑗$. We will clarify this in the main text and highlight that edges serve downstream spatial reasoning and planning.
>
> To address your request for a concrete example, we will add to the appendix a complete JSON‑formatted scene graph corresponding to Fig. 5, including nodes and edges with the above attributes (object phrases/IDs, centroids, bounding boxes, semantic orientations and their language labels; edges with relative translation and size ratio). We will also cross‑reference Fig. 5 where the “JSON‑format Scene Graph” is introduced.
>
> ```json
> {
>   "nodes": [
>     {
>       "id": "obj_1",
>       "phrase": "flashlight",
>       "centroid": [0.32, 0.11, 0.85],
>       "bbox_xyz": [0.28, 0.08, 0.78, 0.36, 0.14, 0.92],
>       "semantic_orientations": [
>         {"illuminate": [0.93, 0.00, 0.36]}
>       ]
>     },
>     {
>       "id": "obj_2",
>       "phrase": "Loopy",
>       "centroid": [0.80, 0.10, 0.85],
>       "bbox_xyz": [0.74, 0.06, 0.78, 0.86, 0.14, 0.92],
>       "semantic_orientations": []
>     }
>   ],
>   "edges": [
>     {
>       "source": "obj_1",
>       "target": "obj_2",
>       "relation": "points_to",
>       "delta_position": [0.48, -0.01, 0.00],
>       "delta_rotation_quat_wxyz": [0.98, 0.00, 0.20, 0.00]
>     }
>   ],
>   "units": {"length": "m", "rotation": "quat(wxyz)", "frame": "world"}
> }
> ```
>
> ## **Q2. Without an affordance map, how does SO / SOFAR decide which part to grasp (e.g., the handle of a knife rather than the blade)?**
>
> Following CoPa, we leverage vision-language models and vision foundation models to identify the target grasping part. After parsing the desired object or part name using a VLM, we employ an open-world grounding module (e.g., Florence-2 or GroundedSAM) to localize the corresponding region and point cloud. Subsequently, the use of SoM, coarse-to-fine grounding, or affordance-based grasping models can further enhance manipulation performance.
>
> Since our primary focus lies in the integration of Semantic Orientation with various robotic tasks, we did not place much emphasis on precise position localization. This is because part-level grounding is sufficient to achieve satisfactory performance for most tasks. Furthermore, **we explore using GraspMolmo as an alternative affordance generator** to replace the Florence-2 + SAM pipeline. This substitution is evaluated across a series of challenging tasks, where it led to performance improvements:
>
> | Task                              | SoFar | SoFar (with graspmolmo) |
> | --------------------------------- | ----- | ----------------------- |
> | Grasp the knife and cut the bread | 4/10  | 6/10                    |
> | Right the fallen wine glass       | 5/10  | 4/10                    |
> | Pour the tea into the cup         | 4/10  | 5/10                    |
> | Open the drawer                   | 5/10  | 6/10                    |
> | Open the door of the cabinet      | 7/10  | 7/10                    |
> | Total success rate                | 50%   | 56%                     |
>
>
> ## **Q3. Coarse six directions limit fine‑grained manipulation (e.g., “pour tea into a cup,” align kettle spout).**
>
> **We acknowledge** that the current data construction pipeline is limited by the use of basic canonical orientations and four oblique views. However, the definition of **Semantic Orientation itself is not restricted to these base directions**, and the predictions produced by PointSO are continuous 3D vectors rather than discrete labels. We anticipate that with more efficient and accurate data construction methods—such as using more views as labels, leveraging world models, using robotic data, or human annotations—it will be possible to handle more complex object orientation estimation tasks.
>
> **Compared to keypoint‑only methods such as ReKep**. Keypoints alone can ensure spatial proximity but may fail to capture orientation. **ReKep relies solely on affordance keypoints generated by DINOv2 for planning and tracking**, which can result in the spout of a teapot being oriented in irrelevant directions, such as to the left or right. Building upon ReKep’s keypoint constraints, **semantic orientation introduces additional constraints** that explicitly account for object orientation, so it will be better than the baseline.

---

> > ### Comment · Reviewer_HWgi · 2025-08-02
> >
> > Since the rebuttal resolved the majority of my concerns, I will maintain my decision to accept.

---

### Official Review · Reviewer_1iDW · 2025-07-06

**Clarity:** 3
**Significance:** 3
**Originality:** 3
**Rating:** 4
**Confidence:** 4

**Summary:**

The authors propose SoFar that integrates object orientation with spatial understanding and manipulation. The authors studies the "semantic orientation" that extends standard 6D pose estimation with language conditions, e.g., "pick up" or "back". Moreover, SoFar integrates a range of visual foundation models and parsing modules and builds a complete parsing of the scene graph. With a strong visual parsing (PointSO and previous visual foundation models) and language reasoning (ChatGPT), SoFar achieves improved results on a range of tasks, especially on ones that require rotation understanding.

**Questions:**

1. Is the PointSO trained on the same set of "semantic orientations" as used in the downstream spatial understanding and manipulation? Does PointSO general to unseen and novel "semantic orientations"?
2. How is the "time cost" in Table 6 estimated?
3. For various modules in Figure 5, how frequent are they executed (e.g., FPS)? What's the running time of various systems?
4. What is the VLM used in SoFar in Table 1?

**Ethical Concerns:**

["NO or VERY MINOR ethics concerns only"]

**Final Justification:**

Rebuttal was convincing. Also read careful response to other reviews. Recommend acceptance.

**Limitations:**

The author mentioned decoupled design of SoFar as a limitation.

**Quality:**

3

**Strengths And Weaknesses:**

**Strengths:**

1. Integrating 6D orientation for spatial understanding and manipulation is an interesting and important research direction.
2. Experiments a thorough and well-designed. Most claims are supported by experimental results and benchmark evaluations demonstrate the advantages of the presented system.

**Weaknesses:**

1. SoFar claims to be an object manipulation model that advances spatial reasoning and orientation understanding. Although explicit 6D orientations as inputs can assist 6D spatial reasoning (as also studied in [A]), it is unclear if the orientation/pose knowledge can truly benefit direct object manipulation, e.g., in a VLA system.
2. The novelty of semantic orientation is limited. It has been studied in prior works: (1) how to align orientations of open-vocabulary objects [A,B], and (2) how to align canonical poses of objaverse objects [B]. Semantic orientation seems to be a trivial extension--e.g., once the 6D orientation of an electric fan is estimated, the "pick up" or "back" direction can be inferred from a simple transformation (with the help of a language model).
3. Is semantic orientation representation necessarily a better option than standard 6D orientation in the SoFar system? Although current PointSO is finetuned with language-orientation paired data that can directly predict "take picture" orientation, using standard 6D orientations in the JSON scene graph should work well too with sufficient finetuning.
4. The spatial reasoning evaluation in Table 6 seems unfair and biased. For models in "VLMs with Spatial Awareness", SoFar seems to be the only model finetuned with 6DoF-related data. Also, besides depth, SoFar also exploits segmentation and pose expert models.
5. Some key technical details are missing and ablation study results are needed, e.g., **Q3** and **Q4**.

[A] 3D-Aware Visual Question Answering about Parts, Poses and Occlusions.
[B] ImageNet3D: Towards General-Purpose Object-Level 3D Understanding.
[C] Orient Anything: Learning Robust Object Orientation Estimation from Rendering 3D Models.

---

> ### Author Rebuttal · Authors · 2025-07-27
>
> We thank the reviewer for the careful reading and constructive feedback. Below, we respond to the main concerns regarding motivation, novelty, fairness of evaluation, and missing details, and we clarify the questions (Q1–Q4).
>
> ## **W1 -- Does semantic orientation truly benefit manipulation?**
>
> **Motivation.** In real-world robotic tasks, many instructions require a nuanced understanding of orientation, particularly in manipulations involving object-object interactions and rich semantic context—for example, inserting a pen into a holder, cutting bread with a knife, or using a flashlight to illuminate. Therefore, **SoFar aims to enhance object manipulation and navigation performance by improving the model’s comprehension of both semantics and orientation**.
>
> **Evidence.** Empirically, SoFar achieves **SOTA performance on the Open6DOR** object rearrangement benchmark—substantially outperforming VLM/VLA baselines—while also showing **SOTA performance on SIMPLER** (both Google Robot and Widow‑X settings). In SIMPLER, SoFar’s **zero-shot** performance surpasses methods trained on a large set of robot trajectories. These results indicate that explicit, language-grounded semantic orientation improves not only spatial reasoning but also downstream manipulation.
>
> ## **W2 -- Novelty of semantic orientation vs. prior art ([A] Po3D‑VQA; [B] ImageNet3D; [C] Orient Anything)**
>
> Our Semantic Orientation maps free‑text descriptions to precise 3D unit vectors in the object’s point‑cloud frame, which planners and controllers can consume directly. It is **designed for manipulation** because VLA systems must translate ambiguous natural language into exact metric actions.
>
> **Key distinctions between semantic orientation and prior work:**
>
> **Po3D‑VQA ([A])**: This work improves VLM's understanding of pose via **VQA (e.g., “Which direction is the bus facing?—Left”)**, i.e., text&vision-in → text-out. Such question-answerings are **not sufficient to execute robotic manipulation**. In contrast, SoFar is text&vision-in → continuous 3D orientation-out, producing vectors that integrate seamlessly with planning and control. Moreover, **[A] only trains with 21 categories of objects** using hand-crafted templates, which can't be seen as open-vocabulary generalization.
>
> **ImageNet3D & OrientAnything ([B], [C])**: These works focus on canonical axes estimation from images. In contrast, our semantic orientation supports open‑world, free‑text queries spanning relative directions, part‑level targets, and **inter‑object interactions (e.g., “USB's plug‑in direction”, “cutting direction of the knife”, “aim the flashlight at Loopy”)**. This functional, instruction‑conditioned grounding is what enables robotic action generation from language.
>
> **Moreover, predicting the semantic meaning of axes from 6D orientation using a language model is non-trivial**, as the language model **cannot access multi-view information** of the object during inference. Inferring the correspondence between axes and semantics (e.g., X-axis <--> pick up, Y-axis <--> cutting) from a single, potentially occluded image is inherently challenging. In contrast, **PointSO is trained on multi-view data and thus achieves superior performance**, as further evidenced by the results in W3.
>
> More importantly, the definition of semantic orientation is not limited to the six canonical directions. The values predicted by PointSO are continuous 3D vectors. During training, we used six canonical views and four oblique views for supervision. Incorporating more views as labels will further enhance directional fitting capability. **In contrast, injecting semantics into standard 6D orientation axes will reduce the problem to constrained six-way classification.** Semantic orientation is not only end-to-end trainable but also exhibits a **higher grain and better scalability**.
>
> **The limitations of prior works motivate the introduction of Semantic Orientation**:
>
> 1. Takes open-world free text as input (e.g., “plug-in direction of a USB”, “cutting direction of a knife”, “point the flashlight at Loopy”), thus jointly generalizing over object categories, parts, and interaction verbs.
> 2. Outputs continuous, metric 3D unit vectors in the object’s point-cloud frame, which can be directly consumed by robotic planners and controllers.
> 3. Is template-free and reference-frame-free, enabling zero-shot generalization without category-specific CAD templates or canonical pose assumptions.
>
> ## **W3 -- Is semantic orientation necessarily better than standard 6D orientation?**
>
> Standard 6D orientation **doesn't encode task semantics** (e.g., “cut”, “plug-in”, “pour out”) and typically **relies on category/template priors** that harm zero-shot generalization. In contrast, semantic orientation is (i) instruction-conditioned & function-aware and (ii) template/frame free. This is crucial when the same object must be manipulated along different functionally meaningful directions (e.g., hold the handle of the mug, pour out the water).
>
> Furthermore, we conduct ablation studies to compare with standard 6-DoF pose estimation. Specifically, we employ FoundationPose & OrientAnything to predict the pose of the manipulated object. Due to the absence of CAD models in FoundationPose, we follow the original protocol by capturing 16 images to construct an implicit representation. After obtaining the predicted 6-DoF pose, we mark it in the image and utilize an **additional VLM (GPT-4o)** to select the desired orientation from the set of standard 6D directions. The real-world manipulation results are as follows:
>
> |Task|SoFar (with PointSO)|SoFar (with FoundationPose + additional VLM)|SoFar (with OrientAnything + additional VLM)
> |-|-|-|-|
> |Grasp the knife and cut the bread|4/10|2/10|2/10|
> |Right the fallen wine glass|5/10|4/10|3/10|
> |Pour the tea into the cup|4/10|1/10|2/10|
> |Open the drawer|5/10|5/10|4/10|
> |Open the door of the cabinet|7/10|6/10|4/10|
> |Total success rate|50%|36%|30%|
>
> It can be seen that our proposed semantic orientation achieves superior performance with fewer VLM calls (less time costs). Furthermore, we would like to clarify that SoFar is an integrated system. In addition to semantic orientation, **our contributions also include the OrienText300K dataset, the Open6DOR V2 benchmark, and a broad suite of downstream tasks spanning cross-embodiment, cross-view, and cross-task generalization**.
>
> Besides, regarding your mention of finetuning standard 6D orientation models on our language-orientation paired dataset, I agree that this approach would be effective, as it essentially aligns 6D orientation with semantics—representing a simplified and discretized form of semantic orientation.
>
> ## **W4 -- Spatial VQA seems unfair; SoFar is the only model finetuned with extra data and also uses extra experts**
>
> First, we divide VQA baselines into “General VLMs” and “VLMs with Spatial Awareness” following their own papers’ claims and training regimes. Many of these already incorporate extra data and off-the-shelf modules (e.g., SpatialBot uses depth API); we report them as-is to respect each method’s intended recipe. We also compare our method with proprietary models such as GPT-4o, which benefit from larger model capacities and broader pretraining corpora.
>
> Besides, we want to clarify that VQA is not the main contribution of our work. SoFar is a robotic manipulation system, and the VQA experiments are conducted to validate the effectiveness of semantic orientation. Our goal is to demonstrate that **incorporating semantic orientation indeed enhances the spatial understanding capabilities of VLMs**, which our results confirm.
>
> ## **Answers to Questions**
>
> ### **Q1. Is PointSO trained on the same set of “semantic orientations” as used downstream? Does it generalize to unseen ones?**
>
> All objects used in our real-world experiments were **not deliberately selected or purchased from the Objaverse dataset**, aligning with the settings of novel objects and in-the-wild generalization. Our OrienText300K dataset contains over 350K diverse objects, with category coverage that matches and exceeds that of LVIS, thus supporting strong category-level generalization. Furthermore, for the evaluations reported in Tables 2 and 3, we ensured fair and accurate testing by separating the training and test sets within the OrienText300K dataset.
>
> ### **Q2. How is the “time cost” in Table 6 estimated?**
>
> The time costs reported in Table 1 and Figure 7 reflect the complete planning duration. For the SoFar model, this includes the processing time of components such as SAM, PointSO, and the VLM. For ReKep, the time costs encompass SAM, DINOv2, and the VLM. For CoPa, they include multiple rounds of SoM, VLM calls, and constraint generation. SoFar demonstrates a clear advantage in time efficiency compared to baselines such as CoPa and ReKep. **CoPa exhibits the highest time cost, primarily due to up to 5 times VLM API calls, whereas SoFar requires only 2 times.** The table below presents the detailed inference breakdown for each method, using a single RTX 4080 16G GPU:
>
> |Methods|VLM|CUDA memory|Planning time costs|
> |-|-|-|-|
> |CoPa|GPT-4o|4375M|20.8s|
> |ReKep|GPT-4o|7020M|15.6s|
> |SoFar-LLaVA|LLaVA|11062M|9.6s|
> |SoFar|GPT-4o|6355M|8.5s|
>
> ### **Q3. Frequency / FPS & running time of modules in Fig. 5?**
>
> All perception & reasoning modules (SAM, PointSO, VLM) are invoked once per planning step. The subsequent low-level execution runs at the robot’s native control frequency (e.g., Franka 1 kHz, UR5e 500 Hz, Unitree Go1 1 kHz), using the planned 6‑DoF trajectory. We will include a table in the appendix with per-module latency and overall FPS for clarity. In brief, the two VLM calls take approximately 5 seconds, while the SAM and PointSO modules require about 1 second, and the Motion Planning module takes around 2 seconds.
>
> ### **Q4. What VLM is used in Table 1?**
>
> SoFar uses GPT‑4o as the agent VLM; SoFar‑LLaVA uses LLaVA-7B as the agent VLM.

---

> > ### Comment · Reviewer_1iDW · 2025-08-05
> >
> > I appreciate the authors' rebuttal, which was convincing. I will raise my score.

---

### Decision · Program_Chairs · 2025-09-17

**Decision:**

Accept (spotlight)

**Comment:**

The paper introduces the concept of semantic orientation, which defines object orientations using language in a reference-frame-free manner, and constructs the OrienText300K dataset to support the task. It further presents the SOFAR framework to incorporate semantic orientation into spatial reasoning and robotic manipulation.

The original concerns from the reviewers include:
- novelty of the semantic orientation: lack of justification for the necessity of semantic orientation beyond standard 6-DoF orientation combined with LLM-based semantic reasoning (1iDW)
- missing details such as the used vlms, edges in the scene graphs, grasping part detection (HWgi, 1iDW)
- lack of computation cost comparison (26tC, MEFz)
- robustness to noisy depths and generalization to complex and fine-grained manipulation tasks (HWgi, MEFz)

The rebuttal adequately addressed these concerns, and the final ratings are 3 Accepts and 1 Borderline Accept. The AC agrees with the reviewers' assessment. The authors should revise the paper accordingly in the final version.